# A mechanosensing mechanism controls plasma membrane shape homeostasis at the nanoscale

Xarxa Quiroga[1,2], Nikhil Walani[3], Andrea Disanza[4], Albert Chavero[5], Alexandra Mittens[1], Francesc Tebar[5], Xavier Trepat[1], Robert G Parton[6], María Isabel Geli[7], Giorgio Scita[4,8], Marino Arroyo[1,9,10], Anabel-Lise Le Roux[1]*, Pere Roca-Cusachs[1,2]*

[1]Institute for Bioengineering of Catalonia, the Barcelona Institute of Technology (BIST), Barcelona, Spain; [2]Departament de Biomedicina, Unitat de Biofísica i Bioenginyeria, Facultat de Medicina i Ciències de la Salut, Universitat de Barcelona, Barcelona, Spain; [3]Department of Applied Mechanics, IIT Delhi, New Delhi, India; [4]IFOM ETS - The AIRC Institute of Molecular Oncology, Milan, Italy; [5]Departament de Biomedicina, Unitat de Biologia Cel·lular, Facultat de Medicina i Ciències de la Salut, Centre de Recerca Biomèdica CELLEX, Institut d'Investigacions Biomèdiques August Pi i Sunyer (IDIBAPS), Universitat de Barcelona, Barcelona, Spain; [6]Institute for Molecular Bioscience and Centre for Microscopy and Microanalysis, University of Queensland, Brisbane, Australia; [7]Institute for Molecular Biology of Barcelona (CSIC), Barcelona, Spain; [8]Department of Oncology and Haemato-Oncology, University of Milan, Milan, Italy; [9]Universitat Politècnica de Catalunya (UPC), Campus Nord, Carrer de Jordi Girona, Barcelona, Spain; [10]Centre Internacional de Mètodes Numèrics en Enginyeria (CIMNE), Barcelona, Spain

*For correspondence:
aleroux@ibecbarcelona.eu (A-LiseLR);
proca@ibecbarcelona.eu (PR-C)

**Abstract** As cells migrate and experience forces from their surroundings, they constantly undergo mechanical deformations which reshape their plasma membrane (PM). To maintain homeostasis, cells need to detect and restore such changes, not only in terms of overall PM area and tension as previously described, but also in terms of local, nanoscale topography. Here, we describe a novel phenomenon, by which cells sense and restore mechanically induced PM nanoscale deformations. We show that cell stretch and subsequent compression reshape the PM in a way that generates local membrane evaginations in the 100 nm scale. These evaginations are recognized by I-BAR proteins, which triggers a burst of actin polymerization mediated by Rac1 and Arp2/3. The actin polymerization burst subsequently re-flattens the evagination, completing the mechanochemical feedback loop. Our results demonstrate a new mechanosensing mechanism for PM shape homeostasis, with potential applicability in different physiological scenarios.

## Editor's evaluation

In this important paper, the authors characterize and model a novel self-organizing circuit used by cells to correct for nanoscale membrane evaginations that form upon a rapid drop in membrane area. The evidence supporting the conclusions is compelling and provides new insights on plasma membrane-cortex coupling. It is of interest to mechanobiologists and biophysicists and could be relevant in processes in developmental biology or in tissues where the cell area changes rapidly, such as in muscles, heart, and lungs.

## Introduction

Cells constantly exchange information with their surroundings, and external inputs are first received by their outermost layer, the plasma membrane (PM). This interface, far from being an inert wall, integrates and transmits incoming stimuli, ultimately impacting cell behavior. In this context, the traditional view of such stimuli as biochemical messengers has now changed to include the concept that physical perturbations are also of major importance (*Apodaca, 2002*; *Beedle et al., 2015*; *Le Roux et al., 2019*). By sensing and responding to physical and biochemical stimuli, one of the main functions of the PM is to adapt to the changes in shape that cells experience as they migrate or are mechanically deformed, in a variety of physiological conditions (*Innocenti, 2018*; *Diz-Muñoz et al., 2016*; *Cheng et al., 2015*; *Li et al., 2022*; *Aragona et al., 2020*; *Gefen, 2011*). To date, research in this area has largely focused on the regulation of PM area and tension, at the level of the whole cell (*Gauthier et al., 2012*; *Pontes et al., 2017*; *Gauthier et al., 2011*). For instance, cell stretch or decrease in medium osmolarity has been commonly used to raise PM tension, unfolding membrane reserves (ruffles, caveolae), inhibiting endocytosis and promoting exocytosis (*Gervásio et al., 2011*; *Dai et al., 1998*; *Riggi et al., 2019*; *Wang and Galli, 2018*; *Lemière et al., 2021*). Conversely, cell exposure to a hypertonic solution or cell compression has been employed to decrease PM tension, leading to an increase on the activity of different endocytic pathways (*Thottacherry et al., 2018*; *Echarri et al., 2019*; *Kosmalska et al., 2015*; *Wang et al., 2011*). These studies have shown that PM tension homeostasis is maintained by regulating PM area through mechanisms like endocytosis, exocytosis, or the assembly and disassembly of PM structures like ruffles and caveolae.

However, changes in cell PM area upon mechanical perturbations are necessarily accompanied by changes in topography at the local scale. This is exemplified by caveolae flattening upon cell stretch (*Sinha et al., 2011*) or creation of PM folds at the sub-µm scale upon cell compression (*Kosmalska et al., 2015*). Curvature also arises when membranes are exposed to either external topographical cues (*Zhao et al., 2017*; *Lou et al., 2019*) or internal pulling by actin filaments (*Galic et al., 2012*; *Itoh et al., 2005*; *Renard et al., 2015*). Thus, to maintain PM homeostasis, cells should be able not only to respond to overall changes in PM tension or area, but also to local changes in PM topography. This requirement is even clearer if one considers recent findings showing that tension does not propagate extensively throughout the whole ensemble of the PM, but dissipates in small areas of less than 5 µm (*Shi et al., 2018*). However, if such local PM shape homeostasis mechanisms exist, and how they operate, is still unknown.

Here, we studied this problem by using as a model the controlled compression of fibroblasts through the application and release of stretch. We show that upon cell compression, bud-shaped PM deformations of negative curvature (evaginations) on the 100 nm scale are formed and enriched by IRSp53, a negative curvature-sensing protein. This creates a local node where specific PM topography is selectively coupled through IRSp53 (and potentially other I-BAR proteins) to activate actin polymerization mediated by Rac1 and Arp2/3. The activation of this cascade flattens the structure, recovering the PM shape to its initial state. Our findings demonstrate that a local mechanosensing mechanism controls PM homeostasis when perturbed through compression.

## Results

### Compression generates dynamic PM evaginations of 100 nm in width

To study how PM topography is regulated, we subjected normal human dermal fibroblasts (NHDFs) transfected with an EGFP-membrane marker to a physiologically relevant 5% equibiaxial stretch by using a custom-made stretch system composed by a PDMS stretchable membrane clamped between two metal rings, as previously described in *Casares et al., 2015* (see Materials and methods). Cell response during and after stretch was monitored by live fluorescence imaging. As previously described, when tensile stress was applied cells increased their area by depleting PM reservoirs, such as ruffles (*Gauthier et al., 2012*; *Kosmalska et al., 2015*). After 3 min, stretch was released, resulting in a compression stimulus. At this point, excess membrane was stored again in folds, visualized as bright fluorescent spots of ≈500 nm (*Figure 1A* and *Video 1*). These spots incorporate approximately 1.5% of PM area (*Figure 2—figure supplement 1A*), and thus store an important fraction of the area modified by cell stretch. As we have previously published (*Kosmalska et al., 2015*), these spots are formed passively by the PM to accommodate compression, analogously to what occurs

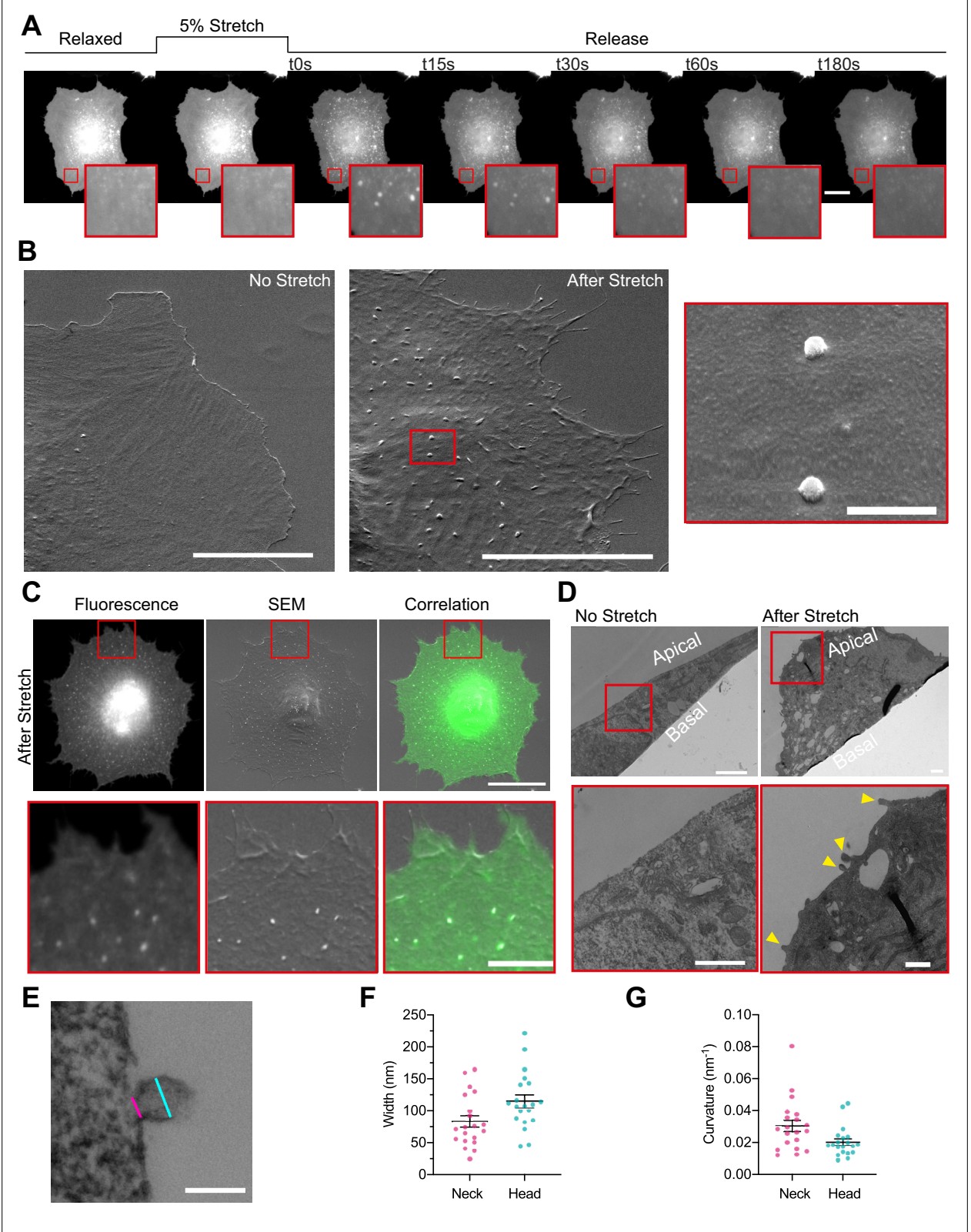

**Figure 1.** Cellular stretch generates plasma membrane (PM) evaginations with a defined curvature. (**A**) Time course images of a normal human dermal fibroblast (NHDF) transfected with EGFP-membrane marker before, during, and after 5% constant stretch application. PM evaginations are seen as bright fluorescent spots after the release of the stretch due to compression of the PM. Scale bar is 20 μm. (**B**) NHDF imaged through scanning electron microscopy (SEM). A non-stretched cell (left), and a cell just after stretch release (right) are shown. Scale bars are 10 μm in main images, 500 nm in

*Figure 1 continued on next page*

*Figure 1 continued*

magnified image (framed in red). (**C**) Correlation between fluorescence and SEM images of a non-stretched and stretched-released NHDF. Matching was achieved by using a patterned substrate together with computational tools for alignment. Scale bar is 20 µm for the main images and 2 µm for the insets. (**D**) Transmission electron microscopy (TEM) images of a non-stretched and a stretched-released NHDF. Yellow arrows in magnified image point at PM evaginations formed at the apical side of the cell. Scale bars are 1 µm for the main images and 500 nm for the insets. (**E**) Detail of an evagination, cyan and magenta lines show evagination's head and neck diameters, respectively. Scale bar is 100 nm. (**F, G**) Corresponding evagination neck and head diameters (**F**) and curvatures (**G**). *N*=22 evaginations from 3 independent experiments. Data show mean ± s.e.m. In A, C, D, and E, red-framed images show a magnification of the areas marked in red in the main image.

The online version of this article includes the following source data for figure 1:

**Source data 1.** Raw data of *Figure 1* graphs and plots.

when compressing synthetic lipid bilayers (*Staykova et al., 2013*). As the diffraction limit of a standard fluorescence microscope lays in the range of 500 nm, we characterized the structure of the compression-generated folds in more detail using electron microscopy. Cells transfected with a PM marker were seeded in a 3D patterned PDMS membrane, stretched and immediately fixed after the release of the stimulus. Next, brightfield and fluorescent images of the 3D pattern and the cells on it were acquired and samples were further processed for scanning electron microscopy (SEM) imaging. Computational alignment tools allowed for correlation between brightfield, fluorescence, and SEM images. De-stretched cells displayed numerous bud-shaped evaginations at their apical PM side that correlated with the bright spots seen by fluorescence (*Figure 1B and C*), showing that the PM bends outward (thereby minimizing friction with the underlying cortex).

Of note, we previously described that cell compression creates not only evaginations at the apical surface but also invaginations in the basal surface (*Kosmalska et al., 2015*). However, most fluorescence spots coincided with evaginations, showing that membrane folds tend to protrude toward the surface offering least resistance (i.e., the media on top of the apical surface, rather than the cytoplasm on top of the basal surface). We thus focused our study on evaginations. To accurately measure the size of these evaginations we moved into transmission electron microscopy (TEM). By comparing non-stretched to stretched-released cells, we observed that the first displayed a homogeneously flat PM, while the second group displayed bud-shaped evaginations on the apical side (*Figure 1D*). Analysis of the shape profile of compression-induced evaginations yielded an average diameter in the neck (cylindrical shape) of 83 nm and of 115 nm in the head (spherical shape), and average curvatures of 0.03 and 0.02 nm$^{-1}$, respectively (*Figure 1E, F and G*).

In cells, passive fold formation is followed by active resorption involving actin cytoskeleton rearrangements, allowing for topography equilibration within minutes (*Kosmalska et al., 2015*). As previously done in mouse embryonic fibroblasts (MEFs) (*Kosmalska et al., 2015*), we plotted the decrease in PM fluorescence at the location of the evagination as a function of time (during 180 s). To assess the effectiveness of resorption, we fitted the fluorescence curve to an exponential equation with a characteristic time scale (*Figure 2A*), quantified by the decay constant in s$^{-1}$. Averaging the decay curves for all cells renders a characteristic resorption time for a specific cell line (*Figure 2B*). Here for NHDF, full reabsorption of evaginations leads to an average decay constant of 0.04 s$^{-1}$ (corresponding to a half-life of about 17 s ensuring a complete return to fluorescent baseline at the end of the experiment).

Altogether, these data indicate that PM compression led primarily to the formation of evaginations of regular size and shape at the apical side, which are immediately resorbed by the cell in an active process to re-equilibrate PM topography and tension.

## Actin is recruited to evaginations

In light of these results, we wondered if the PM evaginations formed upon compression could be sensed by the cell, triggering a mechanism to recover PM shape. Based on previous results

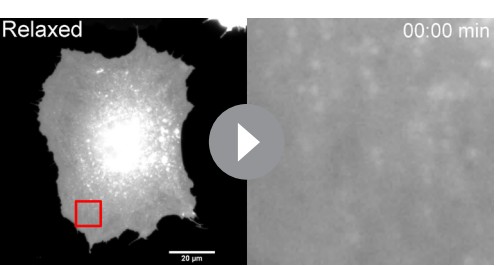

**Video 1.** Time lapse of a normal human dermal fibroblast (NHDF) cell labeled with GFP-membrane before, during, and after stretch application. Images on the right side show a magnification of the areas marked in red on the left side.

https://elifesciences.org/articles/72316/figures#video1

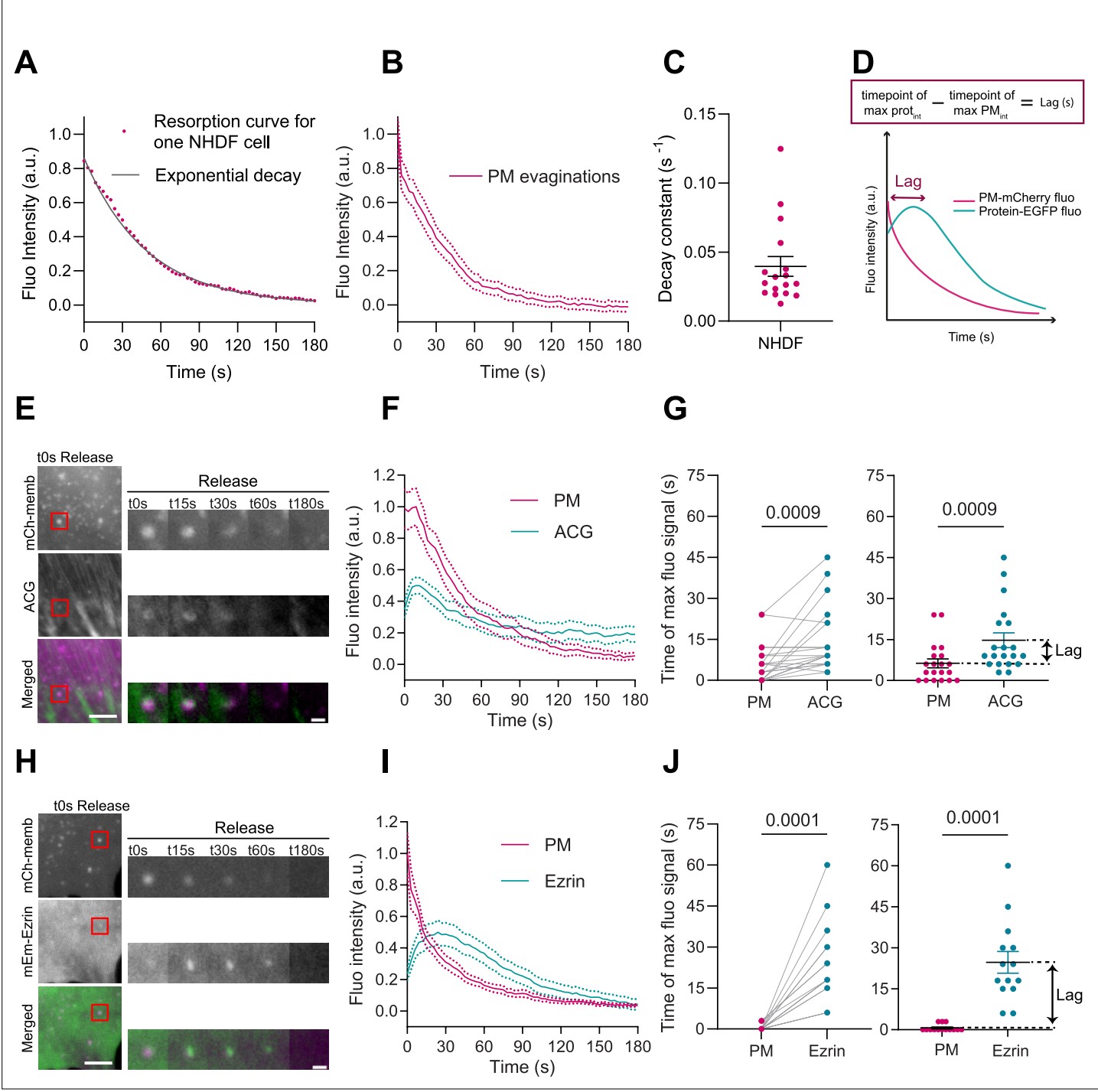

**Figure 2.** Plasma membrane (PM) evaginations trigger local actin recruitment. (**A**) Dynamics of PM evaginations of one example cell after stretch release, quantified as the change in fluorescence of the structure with time (average of 10 evaginations). The black line corresponds to the plot of the corresponding decay curve fit. (**B**) Dynamics of PM evaginations quantified as the change in fluorescence of the structure with time. *N*=17 cells from 3 independent experiments. (**C**) Decay constants extracted from the fits of the PM evaginations dynamics of each cell. *N*=12 cells from 3 independent experiments. (**D**) Definition of lag time. (**E**) Time course images of mCherry-membrane and Actin Chromobody-GFP (ACG) marking PM evaginations in NHDF after stretch release. (**F**) Dynamics of PM evaginations quantified through mCh-membrane or ACG fluorescence markers during stretch release in NHDF. *N*=20 cells from 3 independent experiments. (**G**) Timepoint of maximal fluorescence intensity of PM and ACG (left, paired plot; right, dot plot with mean). Statistical significance was assessed through paired Wilcoxon test. *N*=20 cells from 3 independent experiments. (**H**) Time course images of mCherry-membrane and mEmerald-Ezrin marking PM evaginations in NHDF after the release of the stretch. (**I**) Dynamics of PM evaginations quantified through mCh-membrane and mEmerald-Ezrin fluorescence markers after stretch release in NHDF. *N*=14 cells from 2 independent experiments. (**J**)

*Figure 2 continued on next page*

*Figure 2 continued*

Timepoint of maximal fluorescence intensity of PM and Ezrin markers (left, paired plot; right, dot plot with mean). Statistical significance was assessed through paired Wilcoxon test. *N*=14 cells from 2 independent experiments. Data show mean ± s.e.m.

The online version of this article includes the following source data and figure supplement(s) for figure 2:

**Source data 1.** Raw data of *Figure 1* graphs and plots.

**Figure supplement 1.** Effect of IRSp53 silencing in mouse embryonic fibroblast (MEF) cells.

**Figure supplement 1—source data 1.** Raw data of *Figure 2—figure supplement 1* graphs and plots.

**Figure supplement 1—source data 2.** Figure with the uncropped western blot of *Figure 2—figure supplement 1C*.

**Figure supplement 1—source data 3.** Original file of the full unedited western blot exposed 10 s of *Figure 2—figure supplement 1C*.

**Figure supplement 1—source data 4.** Original file of the full unedited western blot exposed 10 s merged with marker of *Figure 2—figure supplement 1C*.

**Figure supplement 1—source data 5.** Original file of the full unedited western blot exposed 80 s of *Figure 2—figure supplement 1C*.

**Figure supplement 1—source data 6.** Original file of the full unedited western blot exposed 80 s merged with marker of *Figure 2—figure supplement 1C*.

**Figure supplement 1—source data 7.** Figure with the uncropped western blot of *Figure 2—figure supplement 1J*.

**Figure supplement 1—source data 8.** Original file of the full unedited western blot exposed 10 s of *Figure 2—figure supplement 1J*.

**Figure supplement 1—source data 9.** Original file of the full unedited western blot exposed 10 s merged with marker of *Figure 2—figure supplement 1J*.

**Figure supplement 1—source data 10.** Original file of the full unedited western blot exposed 160 s of *Figure 2—figure supplement 1J*.

**Figure supplement 1—source data 11.** Original file of the full unedited western blot exposed 160 s merged with marker of *Figure 2—figure supplement 1J*.

**Figure supplement 1—source data 12.** Original file of the full unedited western blot exposed 240 s of *Figure 2—figure supplement 1J*.

**Figure supplement 1—source data 13.** Original file of the full unedited western blot exposed 240 s merged with marker of *Figure 2—figure supplement 1J*.

showing that actin depolymerization by either latrunculin A or cytochalasin D blocked PM remodeling after stretch (*Kosmalska et al., 2015*), we hypothesized that the first step for recovery might involve reattachment of the evaginated PM to the actin cortex. To explore this idea, we subjected NHDFs to a cycle of stretch and we imaged their response after stretch release. To visualize actin dynamics, cells were co-transfected with a PM marker together with a plasmid expressing an actin nanobody bound to a GFP fluorophore (ACG). As evaginations were being resorbed, actin was recruited to the same spot (*Figure 2E* and *Video 2*). We quantified the fluorescence intensity of both PM and ACG markers and compared the time of maximum intensity in both channels. A difference in times (lag, *Figure 2D*) between any protein marker and the membrane marker indicates that the protein is being recruited subsequently to the formation of the evagination. Actin recruitment was indeed delayed with respect to the membrane (*Figure 2F and G*). This was followed by a decrease in the intensity of both markers that concluded when evaginations were resorbed. This suggests that the PM quickly reattaches to the underlying cortex, which then mediates remodeling of the structure. We noticed that the decay constant quantified for membrane resorption in the cells with ACG overexpression was slightly perturbed (*Figure 2—figure supplement 1B*), which could be due to mechanical interference caused by actin manipulation (*Flores et al., 2019*). We repeated the same experiment by overexpressing the PM-cortex linker Ezrin (*McClatchey, 2014*; *Fritzsche et al., 2014*) and no difference in decay constant was observed under this condition (*Figure 2—figure supplement 1B*). mEmerald-Ezrin also co-localized with evaginations during their resorption (*Figure 2H* and *Video 3*) and fluorescence analysis of PM and Ezrin markers revealed a recruitment of the protein that mimicked, with a delay of 10 s, the one seen with actin (*Figure 2I and J*).

The local appearance of the ACG marker at the evaginated PM suggests a local change in actin architecture, accompanied with reattachment to the cortex. We further explored which molecular machinery can trigger the observed polymerization event. I-BAR domain containing proteins have the ability to sense negative curvature, corresponding to an extruded PM, and to concomitantly recruit actin nucleation promoting factors (NPFs) or even directly bind actin monomers (*Simunovic et al., 2015*). Interestingly, a recent work described how Ezrin needs to act in partnership with the I-BAR protein IRSp53 to enrich in negatively curved membranes (*Tsai et al., 2018*). Moreover, recent studies

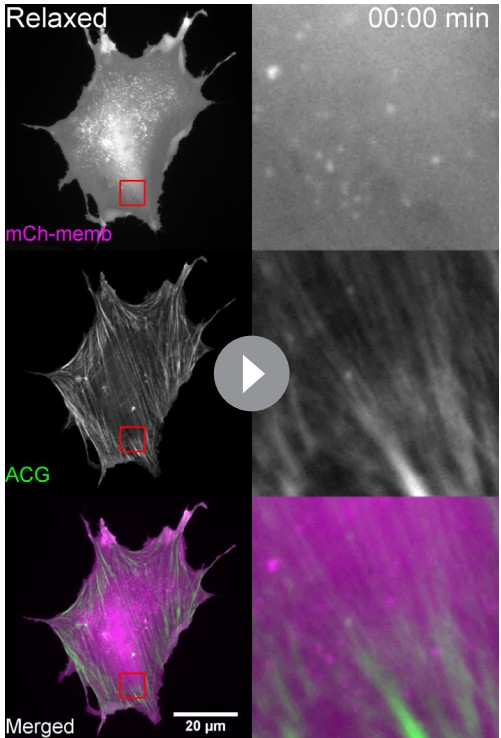

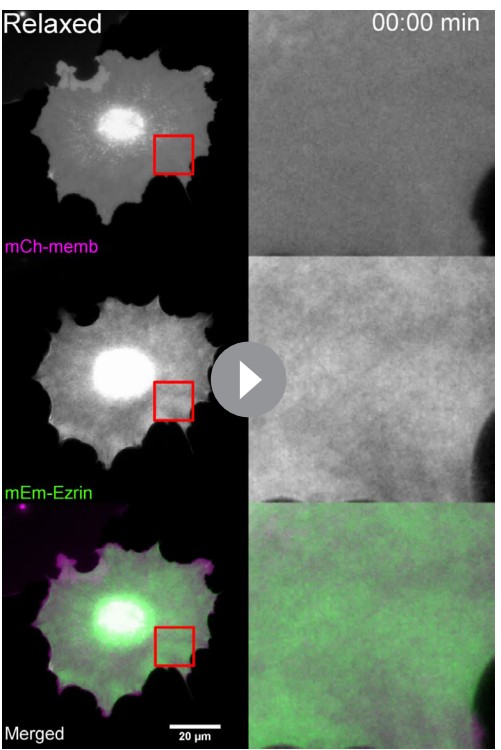

**Video 2.** Time lapse of a normal human dermal fibroblast (NHDF) cell labeled with Actin Chromobody-GFP (ACG) and mCherry-membrane, before, during, and after stretch application. Images on the right side show a magnification of the areas marked in red on the left side.

https://elifesciences.org/articles/72316/figures#video2

**Video 3.** Time lapse of a normal human dermal fibroblast (NHDF) cell labeled with mEmerald-Ezrin and mCherry-membrane, before, during, and after stretch application. Images on the right side show a magnification of the areas marked in red on the left side.

https://elifesciences.org/articles/72316/figures#video3

in vitro and in vivo showed that the I-BAR domain of IRSp53 displays a peak of sorting at evaginations with curvatures of 0.05 nm⁻¹. They also revealed that lower curvature values comparable to the ones obtained by TEM imaging of our evaginations led to a twofold enrichment of this domain with respect to a control membrane marker (*Prévost et al., 2015*; *Breuer et al., 2019*).

Prompted by these observations, we tested if IRSp53 could be the molecular linker between PM shape and actin dynamics in our system. To investigate this possibility, we used isogenic MEFs isolated from IRSp53 null mice (*Disanza et al., 2013*; *Weiss et al., 2009*; *Sawallisch et al., 2009*) that we call IRSp53⁻ᐟ⁻ cells, and compared their decay curves with wild-type (WT) MEF cells (*Figure 2—figure supplement 1C*). No significant differences were observed (*Figure 2—figure supplement 1D, E*).

Despite this result, we examined whether IRSp53 was enriched in the evaginations formed upon stretch release of the PM. For this, we used APEX technology (*Martell et al., 2017*; *Ariotti et al., 2018*) to visualize IRSp53 at PM evaginations using TEM. We co-transfected IRSp53⁻ᐟ⁻ cells with csAPEX2-GBP, a conditionally stable APEX marker bound to a nanobody specifically recognizing GFP (also called GFP-binding protein, GBP), with the EGFP-tagged full length (FL) WT IRSp53 protein (EGFP-IRSP53-FL). As a control, we used a GFP-bound mitochondrial marker (Mito-GFP) instead of EGFP-IRSP53-FL. A strong APEX signal (visible as a darker signal in the TEM image) was observed around the mitochondrial membrane for Mito-GFP-transfected cells (*Figure 2—figure supplement 1F*, top), and at the tip of filipodia for EGFP-IRSp53-FL-transfected cells (*Figure 2—figure supplement 1F*, bottom) as previously described (*Breuer et al., 2019*; *Sathe et al., 2018*). Under these conditions, we analyzed the PM evaginations generated by a stretch-release cycle and found an increase in APEX signal to evaginations only in IRSp53-FL-transfected cells (*Figure 3A*), but not in control

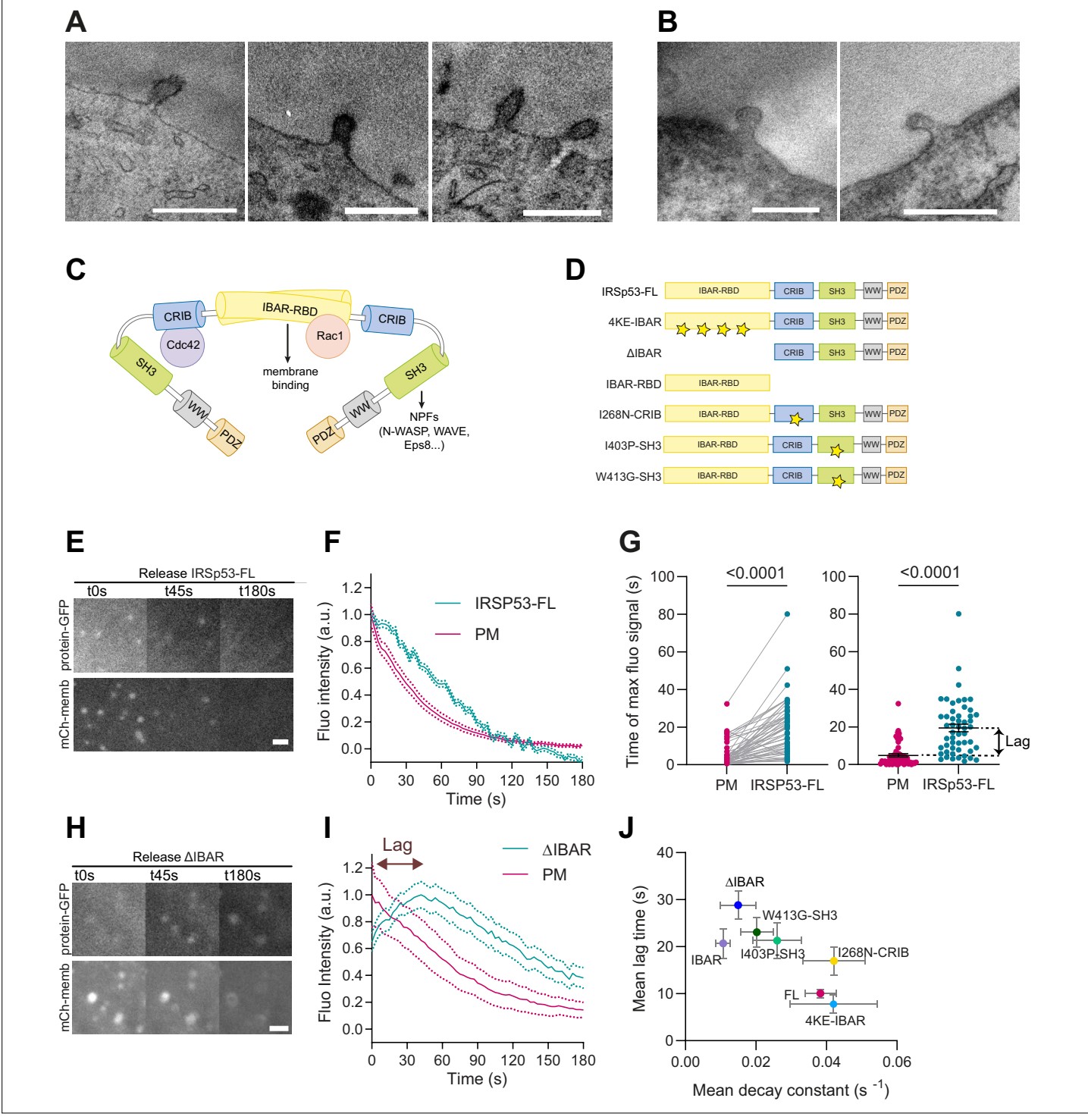

**Figure 3.** I-BAR and SH3 domains of IRSp53 regulate the resorption of plasma membrane (PM) evaginations. (**A, B**) Transmission electron microscopy (TEM) images of PM evaginations coming from cells co-transfected with either APEX-GBP and (**A**) EGFP-IRSP53-FL or (**B**) control condition mito-GFP. APEX staining can be observed at the PM evaginations of EGFP-IRSp53-FL-transfected cells marking IRSp53 position. Scale bars are 500 nm. (**C**) Schematics representing the I-BAR protein IRSp53 and the different molecules interacting with its different domains. (**D**) Schematics of the IRSp53 mutants used in this study. Stars denote the location of mutations impairing the function of the different domains. (**E–F**) Images and dynamics of PM after stretch release of IRSp53$^{-/-R}$ cells transfected with mCh-membrane and FL form of IRSp53 coupled to EGFP. N=53 cells from 12 independent experiments. Scale bar is 2 μm. (**G**) Timepoint of maximal fluorescence intensity of PM and FL form of IRSp53 coupled to EGFP (left, paired plot; right, dot plot with mean). Statistical significance was assessed through paired Wilcoxon test. N=51 cells from 11 independent experiments. (**H, I**) Images and

*Figure 3 continued on next page*

*Figure 3 continued*

dynamics of PM after stretch release of IRSp53$^{-/-R}$ cells transfected with mCh-membrane and ΔI-BAR form of IRSp53 coupled to EGFP. The purple arrow indicates the lag between the PM and IRSp53 signals, that is, the time difference between the peaks of maximum intensity of both markers. *N*=12 cells from 3 independent experiments. Scale bar is 2 μm. (**J**) Time lag of FL or mutated IRSp53 plotted against the decay constant (see *Figure 2—figure supplement 1* for *n* numbers, statistical analyses, and detailed data for each construct). Data show mean ± s.e.m.

The online version of this article includes the following source data and figure supplement(s) for figure 3:

**Source data 1.** Raw data of *Figure 3* graphs and plots.

**Figure supplement 1.** Additional data on IRSp53 mutants.

**Figure supplement 1—source data 1.** Raw data of *Figure 3—figure supplement 1* graphs and plots.

**Figure supplement 2.** Additional data on IRSp53 mutants.

**Figure supplement 2—source data 1.** Raw data of *Figure 3—figure supplement 2* graphs and plots.

mito-GFP-transfected cells (*Figure 3B*). These findings strongly suggest that IRSp53 is recruited to the evaginations.

We thus wondered whether IRSP53 function could be compensated by other I-BAR proteins, and especially IRTKs, a highly homologous member of the I-BAR family proteins. To assess this, we generated IRSP53$^{-/-}$ cells in which IRTKs has been silenced by CRISPR-CAS9 (*Figure 2—figure supplement 1G, H*). We unexpectedly found a slightly faster evagination resorption in the double KO cells (*Figure 2—figure supplement 1I, J*). Potentially, all these results could be due to compensatory overexpression of other I-BAR family members proteins in the different KO cell lines. Consistently, we found that the mRNA levels of the I-BAR proteins IRTKs and MIM were robustly upregulated in the IRSp53$^{-/-}$ cells, while MIM was elevated in the IRSP53$^{-/-}$ IRTKs KO cell line (*Figure 2—figure supplement 1H*). Thus, cells genetically knocked out for IRSp53 adapt to the loss of protein by compensatory elevation of I-BAR family protein, implying that stable KO strategies are not suited to clarify the overall role of I-BAR protein in evagination resorption.

As an alternative strategy, we transiently overexpressed various mutated forms of IRSp53, with potential dominant-negative roles, in cells where the expression of untagged IRSp53 was reconstituted by stable infection (IRSp53$^{-/-R}$ cells). IRSp53 possesses multiple domains with multiple interactors (*Figure 3C*). The I-BAR domain of IRSp53 not only interacts with charged curved membranes, but also possesses a Rac-binding domain binding to activated Rac, and has been described to bundle actin filaments (*Suetsugu et al., 2006b*). IRSp53 also contains an atypical CRIB domain that mediates the interaction with activated Cdc42, but not Rac1 (*Kast et al., 2014*) and, an SH3 domain that can recruit different NPFs, such as WAVE2, N-WASP, the actin regulatory protein Eps8 and VASP (*Scita et al., 2008*). Therefore, we analyzed the effects of various IRSp53 mutants, each affecting a specific domain and impeding a specific interaction, as described in *Table 1* and *Figure 3D*.

We started by overexpressing EGFP-IRSP53-FL protein in IRSp53$^{-/-R}$ cells which colocalized with the PM (*Video 4*). Further confirming IRSP53 recruitment to the invagination, protein fluorescence in the evagination was significantly delayed with respect to the PM marker (*Figure 3E–F*).

Next, we used the fluorescently labeled mutants, which all colocalized with the PM marker (*Figure 3—figure supplement 1A–G*), indicating that multiple protein interaction domains of IRSp53,

**Table 1.** List of IRSp53 mutants used in the experiments.

| Mutant name | Domain affected | Description | Expected effect |
|---|---|---|---|
| IRSP53-4KE | I-BAR | Replace charged lysine by neutral glycine of I-BAR | Decreased electrostatically mediated IRSP53/plasma membrane binding *Suetsugu et al., 2006b*; *Mattila et al., 2007* |
| IRSP53-I268N | CRIB | Loss of function in CRIB | Impaired CDC42 interaction |
| IRSP53-I403P | SH3 | Loss of function in SH3 | Impaired interactions with SH3 interactors, including WAVE2 (*Choi et al., 2005*), VASP, and Eps8 (*Disanza et al., 2013*; *Disanza et al., 2006*) |
| IRSP53-W413G | SH3 | Prevents correct folding of the SH3 domain | |
| I-BAR | SH3+CRIB | Absence of SH3 and CRIB domains | Impaired interactions with SH3 and CRIB interactors |
| ΔI-BAR | I-BAR | Absence of I-BAR | Impaired I-BAR-mediated plasma membrane binding and impaired Rac1 binding |

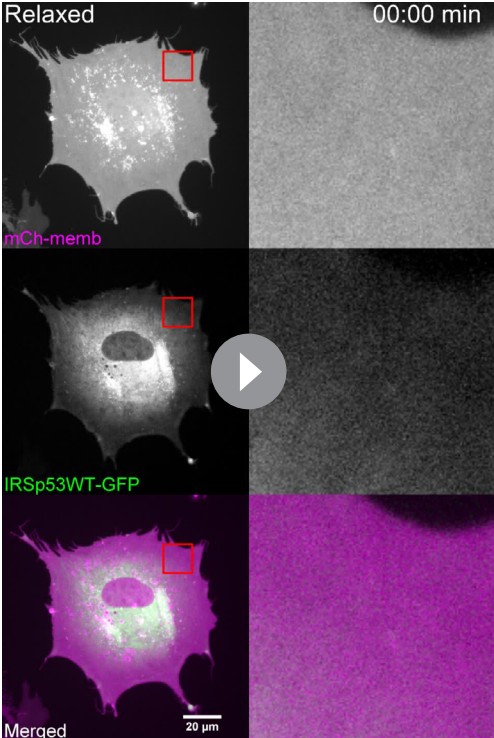

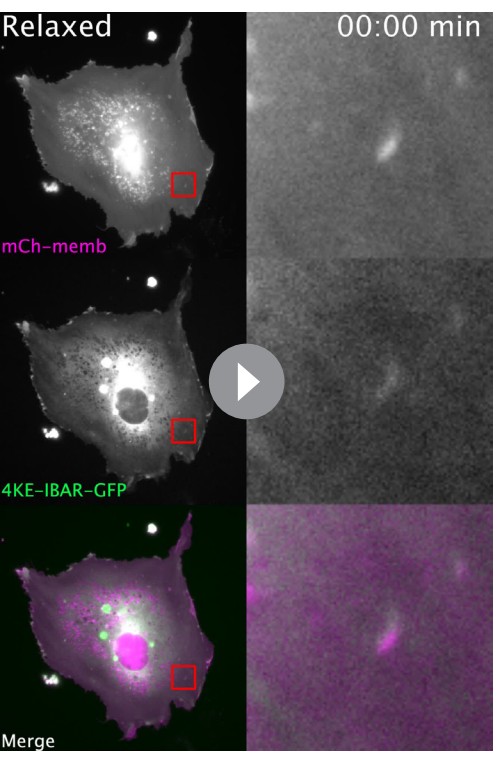

**Video 4.** Time lapse of an IRSp53[-/-R] mouse embryonic fibroblast (MEF) cell, with EGFP-IRSp53-FL overexpression, and labeled with mCherry-membrane, before, during, and after stretch application. Images on the right side show a magnification of the areas marked in red on the left side.

https://elifesciences.org/articles/72316/figures#video4

**Video 5.** Time lapse of an IRSp53[-/-R] mouse embryonic fibroblast (MEF) cell, with EGFP-IRSp53-I268N overexpression, and labeled with mCherry-membrane, before, during, and after stretch application. Images on the right side show a magnification of the areas marked in red on the left side.

https://elifesciences.org/articles/72316/figures#video5

in addition to the I-BAR, mediate the association with the PM, as already suggested in previous studies (*Robens et al., 2010*; *Bisi et al., 2020*). Several of IRSp53 mutants showed slower evagination resorption, and increased lag in recruitment (*Figure 3—figure supplement 1H–L*). Notably, the EGFP-IRSP53-4KE and IRSP53-I268N did not affect PM decay constants and lag times when compared with IRSp53-FL (*Figure 3—figure supplement 2A–D* and *Videos 5 and 6*). In the other cases, however, PM resorption occurred with increased lag times and/or slower decay time, indicating that the over-expression of the mutants affected the process (*Figure 3—figure supplement 2E–H*). Specifically, the point mutant I403P and W413G in the SH3 domain had a small effect on resorption decay constant but significantly increased the lag time. The deletion of the entire I-BAR domain (*Figure 3H, I*) or the overexpression of the I-BAR domain alone significantly impacted, instead, both the resorption decay constant and lag time (also see *Videos 7–10*). Here, we note that the slower recruitment caused by this mutation allowed us to capture the process of IRSp53 recruitment, which was not visible in IRSp53-FL (due to the 5–10 s experimental time required to refocus samples and start imaging after compression). When all mutants were compared, impaired resorption (i.e., decreased decay constant) correlated with delayed recruitment (i.e., increased lag, *Figure 3I and J* and *Figure 3—figure supplement 2C, D, G, H*). The more IRSp53 recruitment was delayed with respect to the PM marker, the less efficient the resorption was.

Since both removal of the I-BAR domain and elevated overexpression of the isolated I-BAR domain had a strong phenotype and Rac1 can bind to the latter, we examined evagination resorption after overexpressing constitutively active (G12V) and dominant negative (T17N) forms of Rac1. Confirming the involvement of Rac1, the expression of Rac1-G12V accelerated evagination resorption significantly whereas Rac1-T17N slowed it down in NHDF (*Figure 3—figure supplement 2I–K*).

IRSp53 has been described to promote Arp2/3-mediated actin polymerization, which is driven by NPFs (WASP or WAVEs), both acting as an upstream (*Connolly et al., 2005*; *Funato et al., 2004*) and

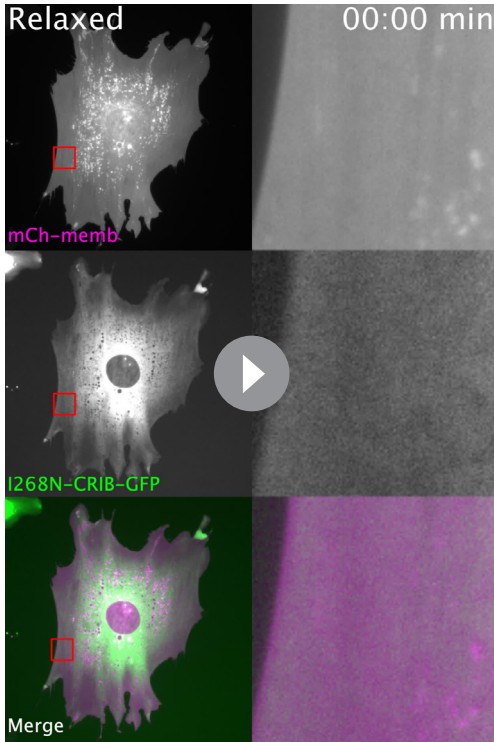

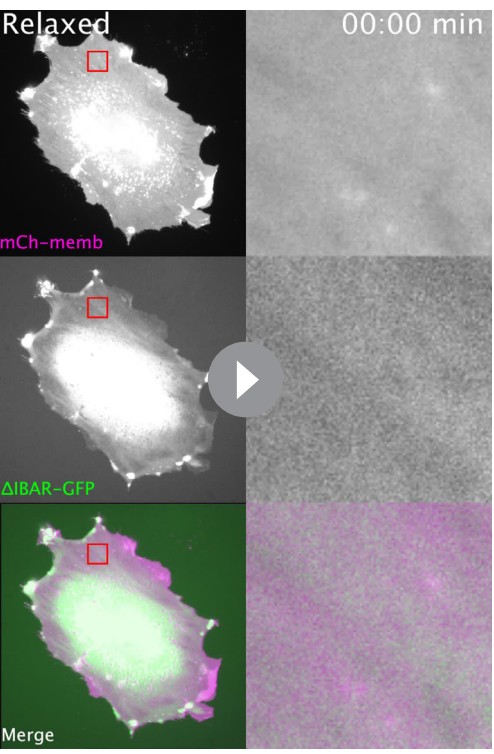

**Video 6.** Time lapse of an IRSp53⁻/⁻ᴿ mouse embryonic fibroblast (MEF) cell, with EGFP-IRSp53-ΔI-BAR overexpression, and labeled with mCherry-membrane, before, during, and after stretch application. Images on the right side show a magnification of the areas marked in red on the left side.

https://elifesciences.org/articles/72316/figures#video6

**Video 7.** Time lapse of an IRSp53⁻/⁻ᴿ mouse embryonic fibroblast (MEF) cell, with EGFP-IRSp53-I408P overexpression, and labeled with mCherry-membrane, before, during, and after stretch application. Images on the right side show a magnification of the areas marked in red on the left side.

https://elifesciences.org/articles/72316/figures#video7

downstream regulator of the small GTPase Rac1. In addition, analysis pointed out a role of the SH3 domain, which is able to recruit WAVEs, but not of the CRIB domain, involved in the recruitment of CDC42. Thus, altogether, these results suggest that IRSp53 by interacting with the WAVE regulatory complex (WRC) and Rac1 might regulate Arp2/3-mediated actin polymerization in evaginations for their resorption, which is what we will subsequently explore.

## The WRC-ARP2/3 molecular machinery mediates the recovery of PM homeostasis after stretch release

Previous work on PM ruffling showed that IRSp53 couples Rac1 to the activation of the WRC, and the subsequent nucleation of branched actin filaments mediated by Arp2/3 (*Abou-Kheir et al., 2008*; *Suetsugu et al., 2006a*; *Goley and Welch, 2006*).

Activation of Arp2/3 downstream of IRSp53 can also be mediated by Cdc42 and N-WASP (*Kast et al., 2014*; *Lim et al., 2008*; *Kurisu and Takenawa, 2009*). Additionally, IRSp53 can coordinate the action of formins mDia1 and mDia2, which might contribute to linear actin polymerization during filopodia formation (*Fujiwara et al., 2000*; *Goh et al., 2012*). Finally, PM reattachment to the actin cortex may also rely on contractile mechanisms mediated by myosin and not only actin polymerization, as in the case of bleb formation (*Charras et al., 2006*). To explore the impact of these mechanisms on evagination resorption after stretch release, we treated IRSp53⁻/⁻ᴿ cells with different inhibitors. First, cell treatment with 10 μM of the N-WASP inhibitor Wiskostatin (*Tsujita et al., 2015*) reduced filopodia number as expected (*Yang et al., 2020*; *Figure 4—figure supplement 1A,B*), but did not modify evagination resorption (*Figure 4A, E, and F* and *Video 11*). Second, treatment with 15 μM of the formin inhibitor SMIFH2 (*Rizvi et al., 2009*) reduced the number of filopodia as expected

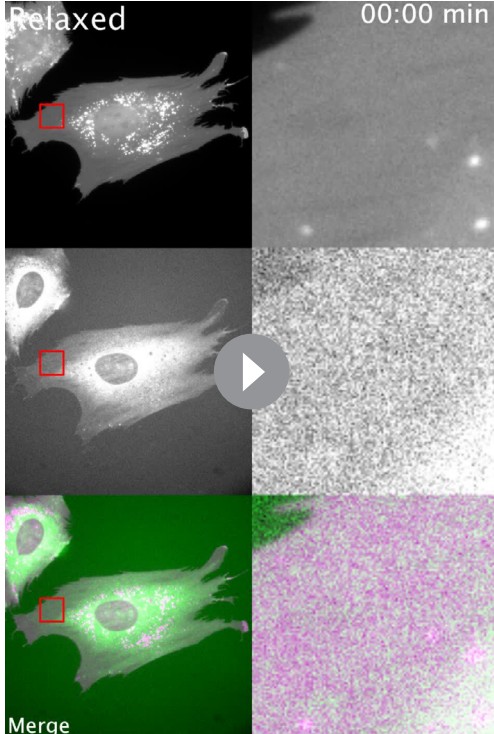

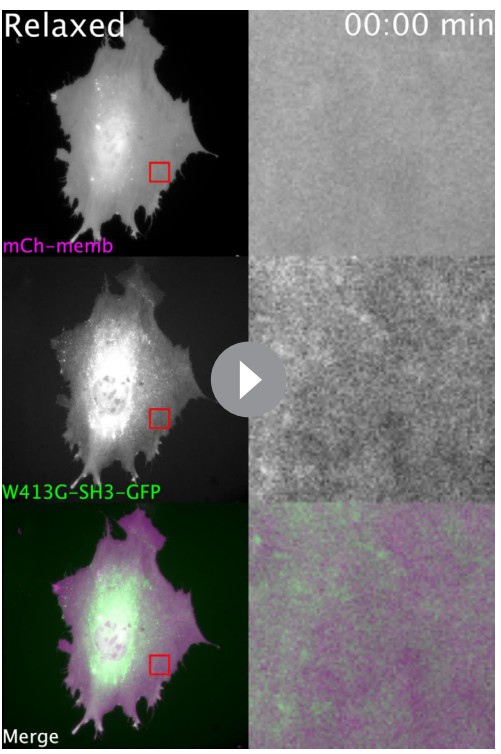

**Video 8.** Time lapse of an IRSp53[-/-R] mouse embryonic fibroblast (MEF) cell, with EGFP-IRSp53-I408P overexpression, and labeled with mCherry-membrane, before, during, and after stretch application. Images on the right side show a magnification of the areas marked in red on the left side.

https://elifesciences.org/articles/72316/figures#video8

**Video 9.** Time lapse of an IRSp53[-/-R] mouse embryonic fibroblast (MEF) cell, with EGFP-IRSp53-W413G overexpression, and labeled with mCherry-membrane, before, during, and after stretch application. Images on the right side show a magnification of the areas marked in red on the left side.

https://elifesciences.org/articles/72316/figures#video9

(*Wakayama et al., 2015*; *Figure 4—figure supplement 1C,D*), but did not affect evagination resorption either (*Figure 4B, G, and H* and *Video 12*). Third, treatment with 10 µM of the myosin II inhibitor Para-nitroblebbistatin (*Képiró et al., 2014*) affected the integrity of stress fibers as expected (*Tojkander et al., 2012*, *Figure 4—figure supplement 1E*) but did not impair evagination resorption (*Figure 4C, I, and J* and *Video 13*). Conversely the treatment with the Arp2/3 inhibitor CK-666 (*Hetrick et al., 2013*) significantly reduced evagination resorption in comparison to DMSO-treated controls (*Figure 4D, K, and L* and *Video 14*).

Thus, Arp2/3 mediates the resorption process, likely through the WRC. To confirm this, we verified WRC localization at the evagination. Upon stretch release, we fixed cells previously co-transfected with GFP-WAVE2 and mCherry-membrane. We then analyzed at high resolution the evaginations showing colocalization of both markers. As evaginations are more difficult to distinguish in fixed samples, we processed the samples for correlative SEM and fluorescence imaging, to visualize apical evaginations using our previous cross-correlation method (*Figure 5A*). With this, we calculated the increase in fluorescence between the evagination and adjacent flat membrane, in both WAVE and membrane markers. This increase was higher for WAVE than for the membrane, indicating a relative enrichment of WAVE at the evagination site (*Figure 5B*).

To further confirm that the local actin polymerization observed was driven by the WRC, we silenced a key component of the complex (*Rottner et al., 2021*), NCKAP1 (Nap1) using siRNA (*Figure 5—figure supplement 1A*). Here again, we employed SEM imaging to visualize evaginations in a control condition (non-targeting siRNA) compared with NCKAP1-siRNA-transfected cells, as co-transfection with the fluorescent marker was incompatible. We then fixed cells after 25 s of stretch release, as resorption at this timepoint is expected to have notably progressed in normal conditions (*Figure 5—figure supplement 1B*). We quantified the number of evaginations per cell PM area in both conditions, segmenting the evaginations using Cell Profiler (*Figure 5—figure supplement 1C, D*). Confirming the role of the

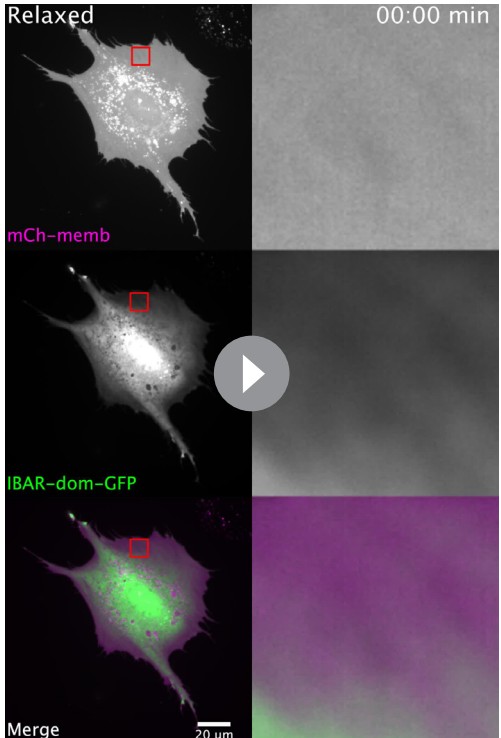

**Video 10.** Time lapse of an IRSp53[-/-R] mouse embryonic fibroblast (MEF) cell, with EGFP-I-BAR overexpression, and labeled with mCherry-membrane, before, during, and after stretch application. Images on the right side show a magnification of the areas marked in red on the left side.

https://elifesciences.org/articles/72316/figures#video10

WRC, the number of evaginations remaining after 25 s of resorption was higher in cells with silenced Nap1 (*Figure 5C–E*). We conclude that evagination resorption upon compression involves the enrichment of I-BAR proteins and WRCs, leading to actin polymerization in a myosin-independent and Arp2/3-dependent manner.

## A mechanical mechanism for actin-mediated evagination flattening

Previous work on IRSp53-mediated actin polymerization describes the extension of out-of-plane protrusions in the form of filopodia or lamellipodia (*Prévost et al., 2015*; *Disanza et al., 2006*; *Connolly et al., 2005*; *Lim et al., 2008*; *Goh et al., 2012*; *Kast and Dominguez, 2019*) as a result of out-of-plane polymerization forces pushing against the PM (*Gov, 2018*). At larger scales, polymerization of an actin cortex retracts and flattens cellular blebs, but this mechanism depends on myosin contractility (*Charras et al., 2006*), and hence is not applicable here. In contrast, our results show a novel flattening rather than protruding response. To propose a plausible mechanism, we developed a theoretical model coupling the PM and the actin cortex (see Materials and methods). We hypothesized that, in addition to out-of-plane forces favoring the extension of evaginations, localized polymerization should also laterally perturb the mechanics of the actin cytoskeleton, resulting in lateral actin flows favoring flattening. We approximated the actin cortex as a flat 2D active gel. In this model, the PM is adhered to the underlying cortex from which it can delaminate, and experiences frictional in-plane forces proportional to relative slippage (*Shi et al., 2018*). This is coupled to our previous model describing curvature sensing of membrane proteins (*Tozzi et al., 2019*). We coarse-grained the signaling pathway triggered by localization of I-BAR protein, such as IRSp53, and leading to actin polymerization through a regulator species (e.g. WRC) with normalized areal density $\psi$, which is recruited beyond a threshold in I-BAR protein enrichment, degraded, and transported by diffusion, with dynamics on time scales comparable to those of actin dynamics. The effect of this regulator is to locally favor actin polymerization by the Arp2/3 complex. Perpendicular to the membrane plane, we modeled polymerization against the tip of the evagination with a force-velocity relation (*Bieling et al., 2016*), according to which the membrane resisting force can slow down, stall, or reverse the extension of the branched network. In the membrane plane, we posited that local polymerization by Arp2/3 biases the competition between a formin-polymerized contractile network component and a branched extensile component (*Chugh and Paluch, 2018*; *Suarez and Kovar, 2016*). We accounted for this lateral mechanical effect of polymerization by locally reducing cortical contractility.

Following the generation of a membrane evagination by buckling-induced delamination (*Kosmalska et al., 2015*; *Staykova et al., 2013*), our model predicts that curvature-sensitive I-BAR proteins become enriched in the evagination within a second (magenta colormap in *Figure 5F*). This leads to recruitment of the regulator species $\psi$ (cyan colormap) resulting in out-of-plane polymerization and a lateral tension gradient in the vicinity of the evagination. Upon contact with the tip of the evagination, out-of-plane polymerization pushes outward and favors extension (yellow arrow). In contrast, the lateral tension gradient induces a centrifugal cortical flow (black arrows), which frictionally drags the inextensible membrane outward favoring flattening. Although the outcome of this competition

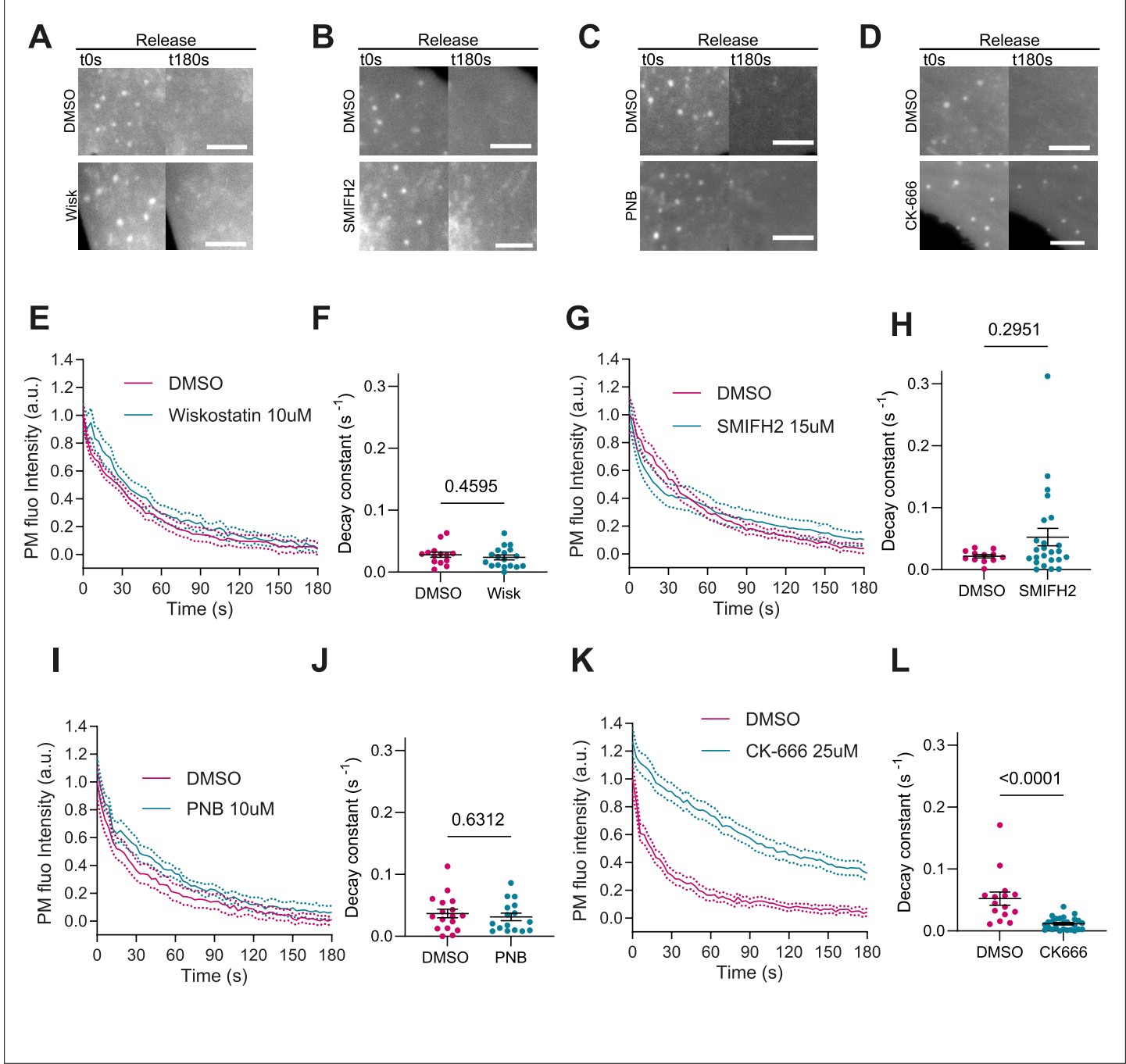

**Figure 4.** Actin polymerization is Arp2/3 dependent, myosin independent. (**A–D**) Images after stretch release of plasma membrane (PM) evaginations, for IRSp53[-/-R] cells treated with either vehicle (DMSO) or 10 μM Wiskostatin, 15 μM SMIFH2, 10 μM PNB, and 25 μM CK-666, respectively. Scale bars are 5 μm. PM is marked with EGFP-membrane. (**E–L**) Corresponding dynamics of PM evaginations and quantification of the decay constants between DMSO-treated control cells and drug-treated cells. Statistical significance was assessed through unpaired t-test for Wiskostatin and PNB against their respective controls, and Mann-Whitney test for CK-666 and SMIFH2 against their respective controls. For Wiskostatin, *N*=18 and 14 cells, SMIFH2, *N*=24 and 12 cells, PNB, *N*=19 and 17 cells, and CK-666, *N*=26 and 15 cells from 3 independent experiments for all cases.

The online version of this article includes the following source data and figure supplement(s) for figure 4:

**Source data 1.** Raw data of *Figure 4* graphs and plots.

**Figure supplement 1.** Controls of drug treatment in IRSp53[-/-R] mouse embryonic fibroblast (MEF).

**Figure supplement 1—source data 1.** Raw data of *Figure 4—figure supplement 1* graphs and plots.

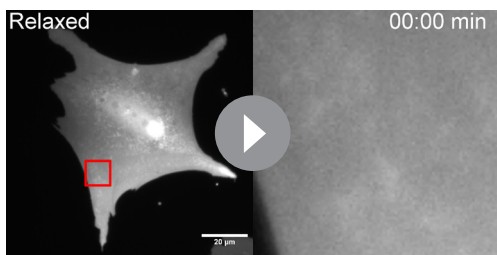

**Video 11.** Time lapse of an IRSp53[-/-R] mouse embryonic fibroblast (MEF) cell treated with 10 μM Wiskostatin and labeled with EGFP membrane, before, during, and after stretch application. Images on the right side show a magnification of the areas marked in red on the left side.

https://elifesciences.org/articles/72316/figures#video11

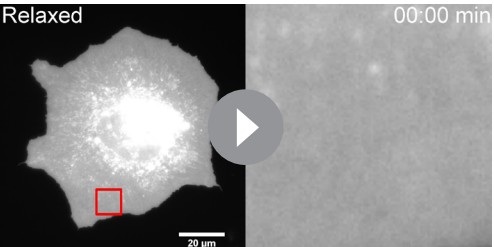

**Video 13.** Time lapse of an IRSp53[-/-R] mouse embryonic fibroblast (MEF) cell treated with 10 μM para-nitroblebbistatin and labeled with EGFP membrane, before, during, and after stretch application. Images on the right side show a magnification of the areas marked in red on the left side.

https://elifesciences.org/articles/72316/figures#video13

depends on model parameters (see discussion of potential scenarios in methods), reasonable estimations lead to flattening of the evagination driven by lateral actin flows and delayed by out-of-plane polymerization. In the absence of curvature, the I-BAR protein-enriched domain dissolves, the regulator species recovers its uniform baseline, and the cortex recovers a quiescent steady state (*Figure 5F and G*). Whereas predicted actin flows occur at a scale well below the diffraction limit and can therefore not be observed experimentally, the predicted relative trends of PM and regulator densities match our experimental observations when comparing PM and actin (*Figure 2G*) or Ezrin (*Figure 2J*). Predictions are also consistent with our observation that evagination resorption is impaired when inhibiting Arp2/3 (*Figure 4L*) or WRC (*Figure 5E*) but not myosin or formin activity (*Figure 4H and J*). Indeed, the mechanism is based on a local gradient in extensile versus contractile behavior around the evagination, and hence it should depend on Arp2/3 (which acts locally at the evagination) and not on formin or myosin, which would regulate overall contractility levels and not specifically local gradients. Thus, our model suggests a chemo-mechanical signaling system that autonomously restores homeostasis of membrane shape and cortical activity.

## Discussion

Our work shows that stretch-compression cycles generate evaginations on the apical PM of the cells with a size on the 100 nm scale, compatible with the sensing range of I-BAR proteins (*Prévost et al., 2015*; *Breuer et al., 2019*). Further, we demonstrate the recognition of this curved templates by the curvature-sensing protein IRSp53. Notably, we show that IRSp53 mediates the resorption rather than formation of membrane curvature, as previously described in the context of filopodia extension (*Disanza et al., 2013*) and HIV budding (*Inamdar et al., 2021*).

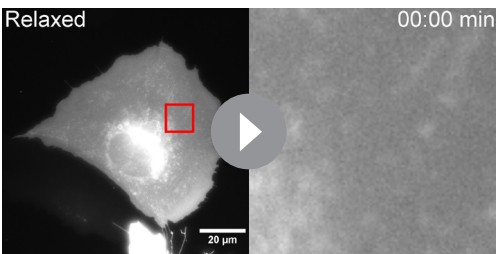

**Video 12.** Time lapse of an IRSp53[-/-R] mouse embryonic fibroblast (MEF) cell treated with 15 μM SMIFH2 and labeled with EGFP membrane, before, during, and after stretch application. Images on the right side show a magnification of the areas marked in red on the left side.

https://elifesciences.org/articles/72316/figures#video12

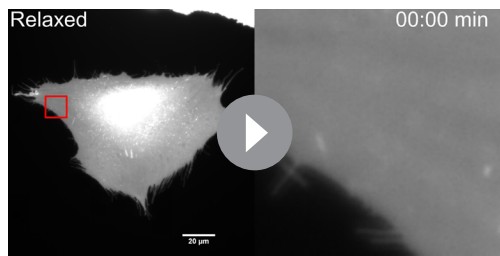

**Video 14.** Time lapse of an IRSp53[-/-R] mouse embryonic fibroblast (MEF) cell treated with 25 μM CK-666 and labeled with EGFP membrane, before, during, and after stretch application. Images on the right side show a magnification of the areas marked in red on the left side.

https://elifesciences.org/articles/72316/figures#video14

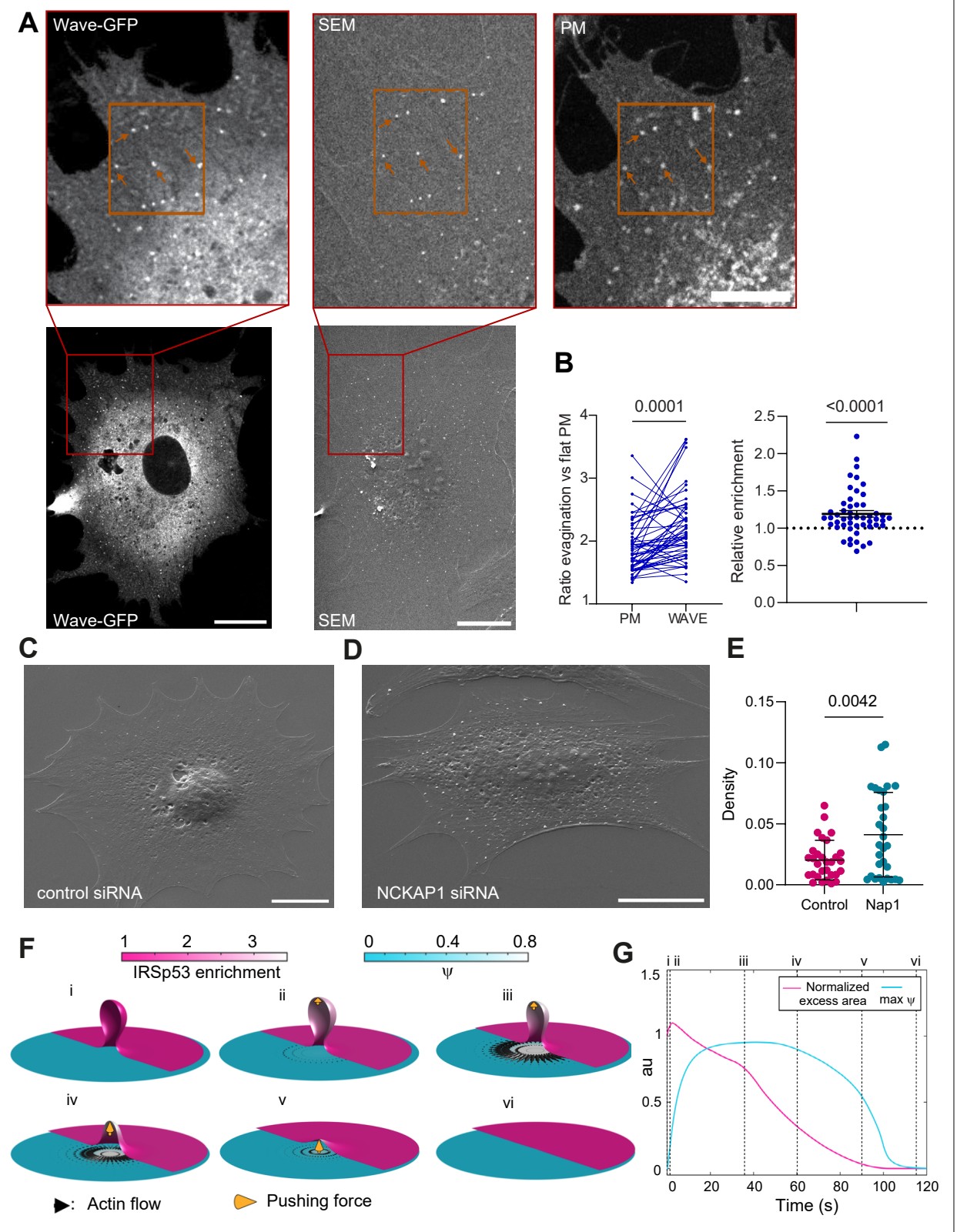

**Figure 5.** The WAVE regulatory complex (WRC) is enriched at the evagination. (**A**) Correlated fluorescence and scanning electron microscopy (SEM) images of plasma membrane (PM) evaginations after stretch release, for IRSp53$^{-/-R}$ cells transfected with mCherry-membrane and WAVE-C-GFP. Scale bars are 10 μm in the full cell image, 5 μm in the inserts (red frame). (**B**) Left: Quantification of the evagination/flat membrane fluorescence ratio in WAVE-GFP and mCherry-membrane fluorescence channels. Right: Corresponding WRC enrichment in the evagination (ratio of ratios). Statistical

*Figure 5 continued on next page*

*Figure 5 continued*

significance was assessed through Wilcoxon and one-sample t-test respectively. *N*=50 evaginations. Data show mean ± s.e.m (**C, D**) SEM images of PM evaginations 25 s after stretch release for IRSp53$^{-/-R}$ cells respectively treated with a non-targeting siRNA or a siRNA against Nap1. Scale bars are 10 μm. (**E**) Quantification of the number of evaginations per μm² of cell membrane area, on cells fixed 25 s after stretch release. The graph shows the density for IRSp53$^{-/-R}$ cells respectively treated with a non-targeting siRNA or a siRNA against Nap1. Statistical significance was assessed through unpaired t-test. *N*=30 cells from 3 independent experiments. Data show mean ± s.e.m (**F**) Dynamics of the model of chemo-mechanical signaling, showing the local enrichment of IRSp53 from a baseline value of 1 (magenta, right side of images) and the concentration of an actin regulator $\psi$ (cyan, left side of images). After the formation of the evagination (**i**), IRSp53 becomes enriched in the bud, which triggers the local increase in the concentration of actin regulator $\psi$ and free growth of branched network which contacts the membrane tip (indicated b yellow arrow) within 2 s (**ii**) followed by build-up of actin regulator $\psi$ over 10 s, thus creating a tension gradient and subsequent centrifugal cortex flow dragging the membrane outward. Because this centrifugal flow flattens the evagination of the inextensible membrane, it is resisted by the pushing force from the branched network as it transits from a closed to an open neck (**iii**), (**iv**). As the bud snaps-open and reduces in curvature, the enrichment of IRSp53 in protrusion is lowered and hence leads to reduction in $\psi$ (**v**). Once planarity is restored, the IRSp53 domain rapidly disassembles, the actin regulator recovers its steady state, and the flow ceases (**vi**). (**G**) Evolution of PM excess area contained in the evagination (where 0 corresponds to a flat membrane patch) and actin regulator concentration $\psi$. Timepoints corresponding to configurations shown in (**F**) are indicated in roman numerals. Excess area is normalized such that 1 corresponds to the initial state and $\psi$ is normalized to a maximum of 1.

The online version of this article includes the following source data and figure supplement(s) for figure 5:

**Source data 1.** Raw data of *Figure 5* graphs and plots.

**Figure supplement 1.** Controls of siRNA treatment with Nap 1 in IRSp53$^{-/-R}$ mouse embryonic fibroblast (MEF).

**Figure supplement 1—source data 1.** Raw data of *Figure 5—figure supplement 1* graphs and plots.

**Figure supplement 2.** Considerations for the model.

Beside IRSp53, other BAR proteins could potentially be involved in evagination resorption. Indeed, the role of IRSp53 is visible through dominant-negative effects of its mutants but not through over-expression or depletion, suggesting that I-BAR proteins with similar function (such as IRTKs) are likely to play compensatory or redundant roles. IRSp53 in complex with WAVE has also been described to localize to saddle curvatures (*Pipathsouk et al., 2021*), which occur at the neck of invaginations. From our APEX imaging, IRSp53 does not appear to localize specifically at the neck of evaginations in our setting, but other saddle- or positive-curvature sensing proteins (such as N-BAR or F-BAR proteins) could potentially localize there. Such localization has been described for instance for the N-BAR protein ArhGAP44 in nascent filopodia (*Galic et al., 2014*). However, ArhGAP44 is expected to inhibit Rac1 activity, which would impair the resorption of the evagination.

Beyond the formation of evaginations, we previously showed that cell compression upon a stretch cycle also triggers endocytosis through the CLIC-GEEC (but not the clathrin-mediated) pathway (*Thottacherry et al., 2018*). In turn, IRSp53 has been described to regulate the CLIC-GEEC endocytic pathway (*Sathe et al., 2018*). However, the pathways involved are different, for two main reasons. First, the I268N-CRIB and 4KE-I-BAR IRSp53 mutants strongly impaired endocytosis (*Sathe et al., 2018*), but did not affect evagination resorption (*Figure 3J*), showing that IRSp53 affects these processes through different mechanisms. Second, as we previously determined, the sites of evaginations and of endocytic buds do not coincide (*Thottacherry et al., 2018*).

Thus, our findings unveil a novel mechanosensing mechanism: upon cell compression, cells are known to use caveolae formation (*Sinha et al., 2011*) and the CLIC-GEEC endocytic pathway (*Thottacherry et al., 2018*) to store material from the PM and recover resting tension. On top of this, we demonstrate a new event at the local scale, which restores PM shape perturbations induced by mechanical stimulation. This event involves the progressive flattening of the PM and not its scission, which would have involved an abrupt loss of evagination fluorescence (and the appearance of fluorescent membrane vesicles) that we never observed in experiments. To achieve PM flattening in response to mechanical perturbations, cells employ the IRSp53-Rac1-Arp2/3 network, which has been well described to polymerize actin in the context of lamellipodia extension or ruffling (*Teodorof et al., 2009*; *Suraneni et al., 2012*). Thus, this involves lamellipodial rather than filopodial formation, which is consistent with our previous observation that IRSp53 does not recruit the formin nucleator mDia (*Disanza et al., 2013*) during initial filopodia formation. In this regard, we describe a novel mechanism, and biophysical framework, in which Arp2/3-mediated actin polymerization can lead to membrane flattening rather than protrusion.

While stretch is often studied separately from subsequent compression provoked by its release (*Gudipaty et al., 2017*; *Massou et al., 2020*; *Chen et al., 2019*), here we put in relevance the coupling between the two at the single cell level. In our experiments, the stretch/compression stimuli occur at the scale of seconds, while subsequent resorption occurs at the scale of minutes. In vivo, similar situations could include for instance the fast compressions of cells embedded in connective tissues (*Zhao et al., 2020*), or the apical expansion and contractions of amnioserosa cells during dorsal closure in *Drosophila* embryos (*Jayasinghe et al., 2013*). Other scenarios could include responses to wounding (*Agha et al., 2011*), contraction in muscle cells, or contractile pulling during fibroblast migration (*Plotnikov et al., 2012*). In all these systems, the potential relevance of our described mechanism remains to be explored. In conclusion, our findings reveal a new mechanosensing mechanism explaining how PM detects physical stimuli at a local, sub-μm scale, and further coordinates a response allowing for quick adaptation to a changing environment.

# Materials and methods
## Cell culture, expression vectors, and reagents

NHDFs were purchased from Lonza (CC-2511) and cultured in DMEM without pyruvate (Thermo Fisher 41965-039) supplemented with 10% FBS (vol/vol), 1% penicillin-streptomycin (vol/vol), and 1% insulin-transferrin-selenium (vol/vol) (Thermo Fisher 41400045). WT MEF2 were derived from IRSp53$^{+/+}$ embryos (*Disanza et al., 2013*; *Weiss et al., 2009*). IRSp53 null MEFs cells were infected with an empty pBABE or a pBABE-IRSp53-FL retroviral vector, generated by G Scita (IFOM ETS, Milan) as previously described (*Disanza et al., 2013*; *Weiss et al., 2009*; *Sawallisch et al., 2009*), leading to cell lines that we note IRSp53$^{-/-}$ and IRSp53$^{-/-R}$. CRISPR/CAS9 method was employed to silence IRTKs in IRSp53$^{-/-}$ background, leading to cell lines that we note IRSp53$^{-/-}$ C/C9IRTKs. All KO cell lines and their controls were authenticated via western blotting as described in the manuscript. All cell lines were regularly tested for mycoplasma contamination and tested negative.

Genome editing by CRISPR/Cas9 was more precisely as follows: IRTKs sgRNA (5'-AAAAGCCTACTA CGACGGCG-3') was subcloned into expression plasmid pSpCas9(BB)–2A-Puro(PX459)V2.0 (Addgene plasmid ID: 62988) and sequences validated by sequencing. 24 hr after transfection, MEFs IRSp53$^{-/-}$ cells were selected for 5 days with culture medium containing 1 μg/ml puromycin (AdipoGen). Clones derived from single cells were obtained by selected population employing serial dilution protocol.

The culture was maintained in DMEM with 20% FBS (vol/vol) supplemented with 1% penicillin-streptomycin (vol/vol) and 1 μ/ml puromycin to selectively maintain cells expressing the selection vector. $CO_2$-independent media (Thermo Fisher 18045088) was used for microscopy imaging and was supplemented with 10 μg/ml of rutin (Sigma R5143) to prevent photobleaching (*Bogdanov et al., 2012*). mCherry, EGFP, and EYFP membrane markers contained a fusion protein consisting in one of the three fluorophores coupled to the 20 last amino acids of neuromodulin which is post-translationally palmitoylated and targets the fluorophore to PM (*Kosmalska et al., 2015*). mEmerald-Ezrin was from Addgene (#54090). pEGFP-C3 Wave2 was kindly provided by Klemens Rottner (Helmholtz Center for Infection Research, Braunschweig). EGFP-IRSp53-FL (*Disanza et al., 2006*; *Bisi et al., 2020*), EGFP-IRSp53-4KE, EGFP-IRSp53-I268N (*Sathe et al., 2018*; *Bisi et al., 2020*), and EGFP-IRSp53-I403P (*Disanza et al., 2013*; *Bisi et al., 2020*) contained isoform 2 of the murine protein either WT or carrying the mentioned mutations in the pC1-EGFP backbone. EGFP-IRSp53-W413G, EGFP-IRSp53-ΔI-BAR, and EGFP-I-BAR (*Disanza et al., 2013*) were created based on the sequence of isoform 4 of the human protein inserted in the pC1-EGFP backbone. A point mutation was included in the SH3, the first 312 amino acids were removed in the case of the ΔI-BAR and the first 250 amino acids were expressed to obtain the I-BAR-domain. The dominant constitutively active Rac1-G12V and the dominant negative Rac1-T17N were described previously (*Soriano-Castell et al., 2017*). Actin was marked using the mammalian expression vector encoding the cytoskeleton marker Actin-VHH fused to either RFP or GFP2 and commercially sold as Actin-Chromobody (Chromotek).

On the day prior to the experiment, cells were transfected by electroporation with the selected plasmids using the Neon Transfection System (Invitrogen) following the protocol provided by the company. CK-666 was purchased from Merck (Ref 182515), SMIFH2 was from Abcam (ab218296), Wiskostatin was bought from Sigma (W2270), and Para-Nitro-Blebbistatin was from Optopharma (DR-N-111). All compounds were diluted in DMSO and conserved according to the manufacturer's

instructions. On the day of the experiment, drugs were diluted in culture media, filtered through a 0.22 µm filter and warmed up to 37°C prior to addition to the culture. Cells were treated with 25 µM of CK-666 for 30 min, 10 µM of PNB for 30–40 min, and 10 µM Wiskostatin or 15 µM SMIFH for 1 hr prior to the experiment.

## siRNA experiment

IRSp53$^{-/-R}$ MEF cells were seeded in a six-well plate (150k cells/well); next day, each well was transfected using lipofectamine (RNAimax, Thermo Fisher) with 10, 25, or 50 nM of control siRNA (Dharmacon, ON-TARGETplus Non-targeting #3) or siRNA against NCKAP1 to silence the Nap1 protein (Dharmacon, ON-TARGETplus Mouse Nackp1 50884). Cells were used 72 hr after transfection occurred.

## Semi-quantitative RT-PCR

### Relative I-BAR protein expression levels in MEFs

Total RNA of WT, IRSP53$^{-/-}$, and IRSP53$^{-/-}$ C/C9IRTKs cells was extracted using the RNeasy Mini kit (QIAGEN) and quantified by NanoDrop to assess both concentration and quality of the samples. One µg of total RNA was subjected to reverse transcription using the High-Capacity cDNA Reverse Transcription Kit from Applied Biosystems. Gene expression was analyzed by using the TaqMan Gene expression Assay (Applied Biosystems). 0.1 ng of cDNA was amplified, in triplicate, in a reaction volume of 25 µl with 10 pmol of each gene-specific primer and the SYBR Green PCR MasterMix (Applied Biosystems). Real-time PCR was performed on the 14 ABI/Prism 7700 Sequence Detector System (PerkinElmer/Applied Biosystems) using a pre-PCR step of 10 min at 95°C, followed by 40 cycles of 15 s at 95°C and 60 s at 60°C. Specificity of the amplified products was confirmed by melting curve analysis (Dissociation Curve; Perkin Elmer/Applied Biosystems) and by 6% PAGE. Preparations with RNA template without reverse transcription were used as negative controls. Samples were amplified with primers for each gene (for details, see the quantitative PCR primer list below) and GAPDH as a housekeeping gene. The cycle threshold Ct values were normalized to the GAPDH curve. PCR experiments were performed in triplicate and standard deviations calculated and displayed as error bars. Primer assay IDs were: Gapdh, mm99999915_g1; Baiap2 (IRSp53), mm00499943_m1; Baiap2l1 (IRTKs), mm00508802_m1; Baiap2l2 (Pinkbar), mm00616958_m1; Mtss1 (MIM), mm00460614_m1; Mtss1l (ABBA), mm01244296_m1.

### Relative Nap1 protein expression levels in MEFs

Total RNA of IRSp53$^{-/-R}$ cells transfected with control siRNA or siRNA against NCKAP1 were isolated with the RNAeasy kit (QIAGEN) or High Pure RNA Isolation Kit (Roche). One µg of total RNA was subjected to reverse transcription using the iScript cDNA Synthesis Kit (Bio-Rad). 0.1 ng of cDNA was amplified, in triplicate, in a reaction volume of 20 µl with 10 pmol of each gene-specific primer and the Fast SYBR Green MasterMix (Thermo Fisher). Real-time PCR was performed on the StepStone-Plus Real-Time PCR System (Applied Biosystems) using a pre-PCR step of 10 min at 95°C, followed by 40 cycles of 15 s at 95°C and 60 s at 60°C. Samples were amplified with primers for each gene (for details, see the quantitative PCR primer list below) and GAPDH as a housekeeping gene. The cycle threshold Ct values were normalized to both GAPDH and Rn18x curves. PCR experiments were performed in triplicate and standard deviations calculated and displayed as error bars. Primer assay IDs were: Nap1 (Sigma, FM1_Nckap1 CATTCGGGGGCTACAATAAAC and BM1_Nckap1 TTAGTGCA GACCGTAAAAAC), GAPDH (FW ATCCTGCACCACCAACTGCT and RV GGGCCATCCACAGTCT TCTG), and Rn18x (FW GCAATTATTCCCCATGAACG and RV GGCCTCACTAAACCATCCAA).

## Western blots

The mouse monoclonal anti-IRSp53 was generated in IFOM (*Disanza et al., 2013*; *Disanza et al., 2006*; *Bisi et al., 2020*). The rabbit polyclonal anti-IRTKs was a gift from Jan Faix Lab (described in *Pokrant et al., 2023*).

## PDMS membrane fabrication

The stretchable PDMS membranes were prepared as described in *Kosmalska et al., 2015*. To produce a patterned support to further obtain patterned-PDMS membranes, PMMA dishes were plasma cleaned for 20 min and warmed up to 95°C for 5 min. After cooling down using a nitrogen gun,

SU 2010 resin was spinned on top of the dish to create a 10 µm layer and prebaked 2.5 min at 95°C. Dishes were then placed on a mask aligner and exposed for 7.5 s in the presence of the designed acetate mask. After post-baking for 3.5 min at 95°C, the pattern was revealed for 1 min and subsequently extensively washed with isopropanol and verified under the microscope. Finally, PMMA dishes were silanized by 30 s plasma cleaning activation followed by 1 hr silane treatment under vacuum. Standard or patterned membranes were mounted on metal rings of our customized stretch system, cleaned, sterilized, and coated with 10 µg/ml fibronectin (Sigma) overnight at 4°C prior to experiments. Patterns were designed as a grid with letters and numbers to allow for correct orientation.

## Stretch and osmolarity experiments

After overnight fibronectin coating, PDMS membranes were quickly washed and 3000 cells were seeded on top and allowed to spread for 45 min to 1 hr in the incubator. Then, rings were mounted on the stretch device coupled to the microscope stage, vacuum was applied for 3 min to stretch the membrane, and then vacuum was released to come back to the initial shape as described in *Kosmalska et al., 2015*. Calibration of the system was done to adjust the vacuum applied to obtain 5% stretch of the PDMS surface.

## SEM experiments

Cells were prepared as explained in the previous section. Right after stretch release, the sample was fixed in 2.5% glutaraldehyde EM grade (Electron Microscopy Sciences 16220) plus 2% PFA (Electron Microscopy Sciences 15710S) diluted in 0.1 M phosphate buffer (PB) at 37°C for 1 hr. Samples were then washed 4× for 10 min in 0.1 M PB and imaged with epifluorescence microscopy as described below to acquire fluorescence images of the cell PM. PDMS membranes were then cut into 1×0.5 cm rectangles in which the pattern was centered. For confocal Airyscan imaging, cells were fixed as abovementionned and PDMS membranes were cut in rectangles and fitted in a glass bottom dish before fluorescent imaging. Samples were subsequently placed on top of 12 mm coverslips for further processing. Dehydration was carried out by soaking samples in increasing ethanol concentrations (50%, 70%, 90%, 96%, and 100%). After this, samples were critical point dried and covered with a thin layer of gold to be imaged.

## TEM experiments

Cells were fixed, washed, and PDMS membranes were cut and mounted as for SEM imaging. After this, samples were postfixed with 1% $OsO_4$ and 0.8% $K_3Fe(CN)_6$ for 1 hr at 4°C in the dark. Next, dehydration in increasing ethanol concentrations (50%, 70%, 90%, 96%, and 100%) was done. Samples were then embedded in increasing concentrations of Pelco EPONATE 12 resin (Pelco 18010) mixed with acetone. 1:3 infiltration was done for 1 hr then 2:2 for 1 hr and finally 3:1 overnight. On the next day, embedding was continued with EPON12 without catalyzer for 3×2 hr washes and then overnight. Last, samples were embedded in EPON12 plus catalyzer DMP-30 (Pelco 18010) for 2×3 hr. To finish, blocks were mounted and polymerized for 48 hr at 60°C. PDMS membrane was next peeled off and ultrathin sections were cut and mounted on grids for imaging.

## APEX labeling for TEM imaging

Two days prior to the experiment, cells were co-transfected by electroporation with mKate2-P2A-APEX2-csGBP (Addgene #108875) and EGFP-IRSp53-FL in a 3:1 ratio, using the Neon Transfection System (Invitrogen) following the protocol provided by the company. Before seeding, cells were sorted for double positive mKate and GFP fluorescence, excluding very high and very low transfection levels. Cells were subsequently seeded and stretched in the same conditions as explained in the stretch experiments section. Right after stretch release, the sample was fixed in 2.5% glutaraldehyde EM grade (Electron Microscopy Sciences 16220) diluted in 0.1 M Cacodylate buffer at 37°C for 10 min, followed by incubation on ice for 50 min in the presence of the fixative. All subsequent steps were performed on ice. The sample was washed three times with cold 0.1 M Cacodylate buffer, and next cut into 1×0.5 cm rectangles containing the fixed cells. Cells were washed for 2 min with a fresh cold 1 mg/ml 3,3'-diaminobenzidine (DAB) (tablets, Sigmafast, D4293) solution in 0.1 M Cacodylate buffer. Cells were immediately incubated with a fresh cold 1 mg/ml DAB solution in cold 0.1 M Cacodylate buffer supplemented with 5.88 mM hydrogen peroxidase (PERDROGEN 30% $H_2O_2$, 31642, Sigma).

The samples were washed three times with cold 0.1 M Cacodylate buffer, and subsequently incubated for 30 min with cold 1% $OsO_4$. Dehydration, resin embedding, and block mounting were done as described in the TEM experiments section.

## Image acquisition

Fluorescence live cells images were acquired with Metamorph software using an upright microscope (Nikon eclipse Ni-U) with a 60× water dipping objective (NIR Apo 60X/WD 2.8, Nikon) and an Orca Flash 4.0 camera (Hamamatsu). Fluorophore emission was collected every 3 s. Cells were imaged in a relaxed state and then for 3 min at 5% stretch, and for 3 min during the release of stretch. Fixed cells images were acquired either in the abovementioned acquisition system except for the experiments related to WAVE protein enrichment. In this case, images were acquired in a Zeiss Airyscan microscope (Zeiss LSM880 inverted confocal microscope objective, using Zeiss ZEN2.3 SP1 FP3 [black, version 14.0.24.201] software and a 63×1.46 NA oil immersion objective). Z-stack of single cells were acquired in full Airyscan mode to visualize the PM and the WAVE-C-GFP protein. SEM images were taken using the xTm Microscope Control software in a NOVA NanoSEM 230 microscope (FEI Company) under the high vacuum mode using ET and TL detectors to acquire high and ultra-high resolution images of the cell surface. TEM samples were observed in a Jeol 1010 microscope (Gatan, Japan) equipped with a tungsten cathode in the CCiTUB EM and Cryomicroscopy Units. Images were acquired at 80 kV with a CCD Megaview 1kx1k.

## Fluorescence analysis and curve fitting

All images used for time course analysis were aligned using the Template Matching plugin from Fiji to correct the drift. To assess the evolution of PM evaginations or the different marked proteins, their fluorescence was quantified. To ensure that we only considered the fluorescence of structures induced by stretch, the analysis was carried out in regions devoid of visible endomembrane structures before the application of stretch. For each evagination, we calculated the integrated fluorescence signal of a small region of interest (ROI) containing the evagination ($I_{evag}$), the integrated fluorescence signal of a neighboring ROI of the same size and devoid of any structures ($I_{PM}$), the integrated fluorescence signal of the entire cell ($I_{cell}$), and the integrated fluorescence signal of a background region of the same size as the cell ($I_{BG}$). Then, the final evagination signal $I_{final}$ was computed as:

$$I_{final} = \frac{\left(I_{evag} - I_{PM}\right)}{\left(I_{cell} - I_{BG}\right)}$$

The numerator of this expression corrects evagination fluorescence so that only the signal coming from the evagination itself and not neighboring PM is quantified. The denominator normalizes by total cell fluorescence, and also accounts for progressive photobleaching. All control curves were normalized to 1 (maximal fluorescence after stretch release) and the rest of the data represented in the same graph were normalized to the control. Exceptionally, Actin and Ezrin curves were normalized to 0.5 (maximal fluorescence after the release of stretch) for visualization purposes. To quantify the degree of resorption of the evaginations, as each experimental data from an evagination could not always be fitted with single exponential decay curve, we adopted the strategy of fitting the average decay curve from 10 evaginations of a single cell as a function of time $t$, using the following equation:

$$I_{final} = I_{max}\, e^{-kt} + I_0$$

We obtained a decay constant $k$ ($s^{-1}$) representative of the resorption capacity of the cell, on which statistical analysis can be performed. Lag time was calculated by identifying the maximum intensity timepoints in the protein and PM channels, and subtracting them to obtain the time between the two events. Lag time was calculated for each individual evagination, and an average of the 10 evaginations of a cell was calculated to obtain a representative lag per cell. This data was used to plot lag time versus decay of each cell. When time of maximum fluorescence was plotted, a similar procedure was followed but maximum fluorescent peak averaged for each cell was displayed instead of lag, in both protein and membrane channels.

The enrichment analysis was performed on the z-stack images acquired in Airyscan mode of cells co-transfected with the mCherry-membrane marker and the WAVE-C-GFP protein. Though sample

preparation was optimized, a single slide of a stack did not capture the best signal for an entire cell. Therefore, the following treatment was performed in both channels: the slice of maximum intensity was identified and a z-projection of the previous and post three slides (sum of seven slides) was performed. Enrichment was calculated using this projection in the following manner: evaginations were identified in the SEM image and 10 evaginations for which both membrane and WAVE protein displayed a fluorescent signal were quantified. We calculated the mean intensity of a small ROI containing the evagination in the protein channel ($I_{evag-Wave}$), and the same ROI was used to calculate that of the membrane channel ($I_{evag-mem}$). An ROI of the same area was placed in a neighboring region devoid of any structures in both channels ($I_{flat-Wave}$ and $I_{flat-mem}$). The ratio of the signal in the flat versus curved membrane was then calculated. Enrichment was defined as the ratio of these ratios.

$$Enrichment = \frac{\dfrac{I_{evag-wave}}{I_{flat-wave}}}{\dfrac{I_{evag-mem}}{I_{flat-mem}}}$$

## Quantification of number and PM area % stored by evaginations

Three regions of different parts of the cell were randomly chosen from every cell at the timepoint t0s (right after the release of stretch) and the number of evaginations was manually counted by comparing the analyzed images with the images of the cell during stretch, to discard PM structures not formed by stretch release. For stored area calculation, the membrane area fraction *mf* contained in evaginations was estimated as:

$$mf = \frac{i_{ze} - i_{zf}}{i_{zf} - bg}$$

where $i_{ze}$ is the average fluorescence intensity of a cell zone (containing evaginations), $i_{zf}$ is the average fluorescence intensity of a neighboring flat patch of membrane (small enough so that it does not contain any evaginations), and *bg* is the average intensity of background. For each cell, this was done for three random regions containing evaginations.

## Fluorescence and SEM correlation

Images of the fixed sample were acquired in fluorescence and brightfield and positions of the imaged cells in the pattern were noted down. Sample was then processed for SEM imaging and the same cells were found by manually following their location on the pattern, and visual verification was done to check for correct matching. Fluorescent and SEM images were then aligned by using the BigWarp plugin on Fiji.

## Statistical analysis

In the case of data following a normal distribution, t-test or ANOVA was done depending on whether there were two or more datasets to compare. For data not following normal distributions, Mann-Whitney or Kruskal-Wallis tests were applied depending on whether there were two or more datasets to test. All data are shown as mean ± SEM. Specific p and *N* values can be found in each one of the graphs shown in the figures.

## Theoretical model

### Modeling a membrane evagination out of an adhered membrane

To understand the physical mechanism leading to the active flattening of membrane evaginations caused by compression of the PM, we focused on a single evagination and described it mathematically under the assumption of axisymmetry. We modeled the membrane as locally inextensible thin sheet with bending rigidity $\kappa = 20\,k_BT$ using the Helfrich model and accounted for the viscous stresses due to membrane shearing with membrane 2D viscosity $\eta_m = 3 \cdot 10^{-3}$ pN · s/µm (*Shi et al., 2018*; *Staykova et al., 2013*; *Arroyo and DeSimone, 2009*).

To model the interaction between the membrane and the cortical gel, we considered an adhesion potential between the membrane and the gel enabling de-cohesion with an adhesion tension of $\gamma = 1.5 \cdot 10^{-5}$ N/m (*Staykova et al., 2013*). We also considered in-plane frictional tractions between

the membrane and the cortex proportional to their relative velocity, $\tau = \mu \left( v_m - v_c \right)$, where $v_m$ is the membrane velocity, $v_c$ is the cortex velocity, and $\mu$ is a friction coefficient, which we took as $\mu = 20$ nN · s/μm³ (**Shi et al., 2018**). See **Figure 5—figure supplement 2A** (top) for an illustration.

We generated evaginations with dimensions comparable to those in our experiment by laterally compressing an adhered membrane patch of radius $R_0$ as discussed in **Staykova et al., 2013**, see **Figure 5F**. We considered $R_0 = 150$ nm, consistent with the typical separation between evaginations (**Figure 1C**). After formation of the evagination, we applied at the boundary of our computational domain the surface tension required to stabilize the evagination, consistent with the long-time stability of such compression-generated evaginations of the PM when cellular activity is abrogated (**Kosmalska et al., 2015**).

## Modeling curvature sensing

We then considered the model in **Tozzi et al., 2019** to capture the interaction between an ensemble of curved proteins (IRSp53) and a membrane. In this model, proteins are described by their area fraction $\phi$. We fixed the chemical potential of such proteins at the boundary of our computational domain, corresponding to a relatively low area fraction of proteins, $\overline{\phi} = 0.05$. We set the saturation coverage to $\phi^{\mathrm{max}} = 0.35$ due to crowding by other species but in our calculations, coverage did not come close to this limit. We considered an effective surface area per dimer of 300 nm². In this model, the curvature energy density of the membrane-protein system is given by $\frac{\kappa}{2} \left( H - C_0 \phi \right)$, where $H$ is the mean curvature and $C_0$ is a parameter combining the intrinsic curvature of proteins and their stiffness (**Tozzi et al., 2019**). We took $C_0 = 3 \cdot 10^{-3}$ nm⁻¹, which lead to curvature sensing but no significant protein-induced membrane reshaping. With a protein diffusivity of 0.1 μm²/s, we obtained protein enrichments on the evagination of about threefold within 0.5 s.

## Modeling signaling of localized actin polymerization downstream of curvature

Branched filaments, on the other hand, are assumed to grow out-of-plane of the gel toward the membrane protrusion. In vitro studies combining TIRF and AFM (**Bieling et al., 2016**) have shown how Arp2/3-mediated branched network grows and densifies when it encounters resistance, leading to a growth velocity that exponentially decays with pressure or stress felt by the growing ends. In this study, we assume that the net effect of pressure applied by the filaments can be accounted by a point load (a product of pressure and the area commensurate to opening at the neck of budded domain) that is directed away from gel and toward the evagination at the membrane tip yielding the constitutive law for growth of filamentous actin network as shown in **Figure 5—figure supplement 2A**.

To model in a coarse-grained manner the signaling pathway triggered by IRSp53 localization and leading to actin polymerization, we considered a regulator species given by a normalized surface density $\psi$, which is produced with a rate depending on IRSp53 enrichment and given by $r \left( \frac{\phi}{\overline{\phi}} \right) = k_1 \min \left\{ \left\langle \frac{\phi}{\overline{\phi}} - e_t \right\rangle ; e_s - e_t \right\}$, where $e_t$ is a threshold IRSp53 enrichment for signaling, $e_s$ is an enrichment saturation threshold beyond which the production of $\Psi$ saturates, and $\langle a \rangle$ is 0 if $a < 0$ and $a$ otherwise. The baseline area fraction of IRSp53 is denoted by $\overline{\phi}$, which is also prescribed as a boundary condition with value $\overline{\phi} = 0.05$. We considered $e_t = 2$, $e_s = 3$, and $k_1 = 1$ s⁻¹. This regulator is degraded with rate $k_2 \psi$, with $k_2 = 1$ s⁻¹ and diffuses with an effective diffusivity of $D = 0.1 \cdot 10^{-3}$ μm²/s, much smaller than that of membrane proteins since the regulator is viewed as an actin-binding species. In polar coordinates, the governing equation for the transport of this regulator is thus

$$\frac{\partial \psi}{\partial t} = \frac{1}{r} \frac{\partial}{\partial r} \left( r D \frac{\partial \psi}{\partial r} \right) + r(\phi/\overline{\phi}) - k_2 \psi \tag{1}$$

This equation results in a region enriched with $\psi$, co-localizing with the evagination, and reaching a maximum value of about 1 within about 10 s, comparable to the typical times of actin dynamics. Not being a detailed description of a specific network, the details of this model for $\psi$ are not essential. The key points are that the production of $\psi$ is triggered by IRSp53 enrichment, and that $k_1$, $k_2$, and $D$ are such that over the time scales of actin dynamics (significantly slower than those of IRSp53 enrichment), a region of high $\psi$ develops close to the evagination.

## Modeling the mechanical effect of localized actin polymerization

The effect of regulator $\psi$ is to locally favor actin polymerization by the Arp2/3 complex. We simplified the mechanical effect of localized actin polymerization by splitting the in-plane and out-of-plane effects. In-plane, we modeled the actin cytoskeleton adjacent to the membrane as a 2D planar active gel. Out-of-plane, we modeled the growing protrusion of Arp2/3-mediated branched network against the membrane as an effective filament, whose width is commensurate to the neck of the evagination and whose growth/retraction is force-dependent. These two effects of polymerization compete to determine the dynamics of membrane protrusions as elaborated below.

### Out-of-plane effect

A localized branched network is assumed to grow out-of-plane of the gel toward the membrane protrusion. In vitro studies combining TIRF and AFM (*Bieling et al., 2016*) have shown how Arp2/3-mediated branched network grows and densifies when it encounters resistance, leading to a growth velocity that exponentially decays with pressure felt by the growing ends. Here, we assume that the net effect of forces applied by the filaments can be accounted by a point force (the product of pressure and the area commensurate to the opening at the neck of budded domain) directed away from gel and toward the evagination at the membrane tip. We consider for this idealized point representation of the pushing force a relation between growth velocity and force shown in *Figure 5—figure supplement 2A* (bottom) based on data by *Bieling et al., 2016*, with zero-force polymerization velocity of 100 nm/s and stall force of about 2 pN. Thus, the out-of-plane effect of the growing membrane is a mixed force-velocity (Robin) boundary condition for the membrane tip where force acting ($f$) is related to the vertical velocity of this point $v_t$ according to the following fit of data by *Bieling et al., 2016*

$$v_t = v_t^0 \left( e^{-\beta f} - e^{-\beta f_s^0} \right) \tag{2}$$

Here, $v_t^0 = 90$ nm/s * $\langle \max(\psi) - \epsilon \rangle$ is approximately the freely growing speed of the branched network when $\psi$ is accumulated beyond a threshold $\epsilon = 10^{-2}$, $\beta \approx 2.4$ pN$^{-1}$ specifies the force sensitivity and $f_s^0 \approx 1.82$ pN is the stall force obtained as product of stall pressure of ~1250 Pa times the circular area around the tip of membrane that is commensurate to the neck opening of ~1500 nm$^2$. For forces larger than stall force, the network retracts at a rate close to an asymptotic velocity of 1 nm/s.

### In-plane effect

Moving to the in-plane effect, the cortex can be viewed as a composite system of interpenetrating actin networks, one polymerized by formins leading to linear filaments and producing contractile forces through the action of myosins and other crosslinkers, and one polymerized by the Arp2/3 complex, with a branched architecture and producing extensile forces by polymerization (*Chugh and Paluch, 2018*). Combining these two effects, the net active force generation in the actin cortex is generally contractile. These two networks compete for actin monomers (*Suarez and Kovar, 2016*), and hence a local enrichment in the regulator leading to enhanced polymerization of the branched network should bias this competition and locally lower contractility in the vicinity of the evagination. In turn, the resulting contractility gradient should generate an in-plane centrifugal cortical flow, which if large enough, might drag the membrane outward due to frictional forces and actively flatten the evagination.

To model the actin flow, we considered simple active gel model where the cortical velocity $v_c$ is obtained by force balance between viscous and active forces in the cortex, and given by

$$0 = 2\eta_c \left[ -\frac{1}{r}\frac{\partial}{\partial r}\left( \frac{r\partial v_c}{\partial r} \right) \right] + \frac{\partial \sigma^a}{\partial r} \tag{3}$$

where $\eta_c$ is the viscosity of the cortex and $\sigma^a(\psi)$ is the active tension, which we assume to be a function of the regulator $\psi$. We note that we neglect in the equation above the force caused by friction between the membrane and the cortex as they slip past each other. This is justified because the hydrodynamic length for the cortex is in the order of microns and above, and hence in the smaller length-scales considered here viscosity dominates over friction. In our calculations, we took

$\sigma^a\left(\psi\right) = \sigma^0\left(1 - \frac{\psi}{2}\right)$, so that active tension is approximately halved near the evagination when the normalized regulator density $\psi$ reaches about 1 and is equal to $\sigma^0$ far away from it. As boundary conditions, we considered $v_c\left(0\right) = 0$ consistent with polar symmetry and $\partial\frac{v_c}{\partial r}\left(R\right) = 0$, so that at $r = R$ the stress at the gel is $\sigma^0$. We chose $\sigma^0/\eta_c$ so that the resulting cortical velocities due to gradients in active tension gradients were of about 0.1 μm/s, comparable to the typical actin velocities due to polymerization in the lamellipodium (*Maiuri et al., 2015*).

## Representative results

### Complete model

The formation of a membrane evagination by buckling-induced delamination of the fluid membrane triggered in this model a sequence of chemo-mechanical events restoring autonomously homeostasis of membrane shape and the rest of signaling species involved.

Indeed, within a few seconds, IRSp53 became enriched in the evagination by curvature sensing. As the concentration of BAR protein builds up, it progressively builds up in its turn the actin regulator ($\psi$) in the vicinity of evagination over a time scale of ~10 s. Because the force-free polymerization velocity $v_t^0$ perpendicular to the membrane depends on actin regulator, as soon as $\psi$ becomes non-zero, branched Arp2/3 network grows freely until it contacts with the tip of membrane at time ~1 s, when it starts pushing on it. Concomittantly, the gradient in $\psi$ generates a lateral gradient in active tension $\sigma^a$, which in turn drives a centrifugal cortical flow. This flow causes a centrifugal friction traction on the membrane and hence favors outward flow. However, this mechanism competes with the out-of-plane force on the evagination by localized polymerization. The outcome of this competing mechanisms depends on their relative strength.

For the parameters considered here, the pushing force normal to the membrane plane (orange arrow in *Figure 5F* and orange plot in *Figure 5—figure supplement 2B*.) quickly reached the stall force $f_s^0 \sim 2$ pN, marking the onset of normal retraction of the branched network (instant ii). This normal force continues to increase as the gradient in $\psi$ and active tension is established (iii and iv). In these stages, it rises an order of magnitude larger than the stall force, enabling rapid retraction of the protrusion driven by the centrifugal action flows. Thus, friction-mediated forces from lateral gradient in active tension overcome the pushing force from filaments to iron out the evagination. The retraction velocity of the branched network under force dictates the time scale of flattening. An evagination of 100 nm reducing at a speed of ~1 nm/s flattens in 100 s, corresponding to the typical time scale of flattening. Following membrane flattening, the IRSp53 domain is rapidly dissolved and according to *Equation 1*, $\psi$ drops to zero everywhere, eventually stopping the cortical flow and thus recovering a homeostatic state with a planar membrane and a quiescent cortex.

Our model is consistent with the fact that myosin inhibition does not affect the resorption process. Indeed, myosin inhibition should lower the baseline active tension, $\sigma_0$, but should not change the fact that localized polymerization would locally induce and extensile stress, and hence establish a tension gradient and a centrifugal actin flow.

### Alternative scenarios

We confront this situation with two alternative scenarios. First, we neglect the out-of-plane force by actin polymerization. This is imposed by setting $v_t^0 = 0$ in *Equation 2*. The initial stages following the formation of the evagination progress similarly as before, *Figure 5—figure supplement 2C* (A), with the enrichment of IRSp53 on the bud (ii), followed by build-up of actin regulator $\psi$ (iii), leading to a gradient in contractility and the centrifugal flow dragging the membrane to flatten the evagination (iv and v), and recover membrane-cortex homeostasis. However, without the resistance from branched actin network, the resorption of the evagination proceeds by a snapping event characteristic of bud formation/flattening (*Tozzi et al., 2019*), which takes place over ~10 s, in comparison to the ~100 s observed in experiments and predicted by the previous model. Hence, in this scenario, the system also recovers homeostasis, yet the resorption dynamics are unrealistically abrupt.

Next, we consider a second alternative scenario in which we neglect the lateral cortical flow and just retain the out-of-plane force, considering growth velocity of branched network decoupled from presence of actin regulator species such that $v_t^0 = 90$ nm/s and zero lateral actin flow. We find that the pushing force slightly increases the size of a protrusion leading to a stable bud, *Figure 5—figure*

*supplement 2C* (B). Hence, the polymerization force is not able to overcome membrane tension and recruit additional membrane area. However, actin polymerization is known to drive elongation of membrane extension, for example endocytic buds, filopodia, or lamellipodia. It is reasonable to expect that the force-velocity relation of the growing network depends on the composition of the actin cytoskeleton. For instance, in vitro studies and theory suggest that in the presence of bundling agents, branched networks more easily form membrane tubes and increase their stall force (*Liu et al., 2008*; *Tsekouras et al., 2011*). Accordingly, we found that increasing the stall force by an order of magnitude drives the evagination into tubular domains as shown in *Figure 5—figure supplement 2C*.

## Summary

In summary, these results suggest that localized signaling induced by curvature sensing triggers localized polymerization of branched actin by Arp2/3. This localized polymerization has two opposite mechanical effects as it generates (1) out-of-plane polymerization forces driving further extension of the evagination and (2) a lateral contractility gradient driving centrifugal actin flow, which drags and tends to flatten the evagination. This competition may lead to different outcomes depending on the relative strength. If (1) is negligible, then the centrifugal actin flow irons out the evagination very abruptly. If (2) is negligible, then polymerization forces may extend the evagination further. In an intermediate regime that agrees with observations, lateral actin flow controls the outcome, that is recovery of membrane cortex homeostasis, but out-of-plane polymerization forces affect the dynamics by delaying flattening.

## Acknowledgements

We thank V González-Tarragó for assistance with the stretch system and statistics analysis, L Rosetti for support with Fiji scripts, I Granero for helping with CellProfiler pipelines, J Oliver De La Cruz for help with RT-PCR, N Castro, S Usieto, and A Menéndez for technical assistance and the members of the PR-C and XT laboratories for technical assistance and discussions. We would also like to acknowledge the support given by the Unitat de Criomicroscòpia Electrònica TEM/SEM (Centres Científics i Tecnològics de la Universitat de Barcelona, CCiTUB),the MicroFabSpace and Microscopy Characterization Facility, Unit 7 of ICTS 'NANBIOSIS' from CIBER-BBN at IBEC, and the V Dall'Olio and L Tizzoni qRT-PCR service – Cogentech SRL facility. Funding: Spanish Ministry of Science and Innovation (PGC2018-099645-B-I00 to XT, PID2019-110298GB-I00 to PR-C and BFU2016-79916-P to XQ). European Commission (H2020-FETPROACT-01-2016-731957) European Research Council (Adv-883739 to XT). Generalitat de Catalunya (2021 SGR 01425 to XT and PR-C). The prize 'ICREA Academia' for excellence in research to PR-Cand MA. Fundació la Marató de TV3 (201936-30-31). Obra Social 'La Caixa' (agreement LCF/PR/HR20/52400004). IBEC is recipient of a Severo Ochoa Award of Excellence from the MINCIN. AC was supported by a FPU fellowship from Ministerio de Educación, Cultura y Deporte (Spain). Grant BFU2015-66785-P from the Ministerio de Economía y Competitividad (Spain) to FT. Associazione Italiana per la Ricerca sul Cancro AIRC-IG 18621 and 5XMille22759 to GS. The Italian Ministry of University and Scientific Research (PRIN 2017-Prot. 2017HWTP2K to GS).

# Additional information

### Competing interests

María Isabel Geli: Reviewing editor, *eLife*. The other authors declare that no competing interests exist.

### Funding

| Funder | Grant reference number | Author |
| --- | --- | --- |
| Ministerio de Ciencia e Innovación | PID2019-110298GB-I00 | Pere Roca-Cusachs |
| European Commission | H2020-FETPROACT-01-2016-731957 | Xavier Trepat |
| Generalitat de Catalunya | 2021 SGR 01425 | Xavier Trepat |

| Funder | Grant reference number | Author |
| --- | --- | --- |
| Fundació la Marató de TV3 | 201936-30-31 | Pere Roca-Cusachs |
| 'la Caixa' Foundation | LCF/PR/HR20/52400004 | Pere Roca-Cusachs Xavier Trepat |
| Ministerio de Ciencia e Innovación | BFU2015-66785-P | Francesc Tebar |
| Associazione Italiana per la Ricerca sul Cancro | AIRC-IG 18621 and 1311 5XMille22759 | Giorgio Scita |
| italian ministry of university | PRIN 2017-Prot. 1313 2017HWTP2K | Giorgio Scita |
| European Research Council | Adv-883739 | Xavier Trepat |
| Institució Catalana de Recerca i Estudis Avançats | ICREA Acadèmia Prize | Pere Roca-Cusachs |
| Ministerio de Ciencia e Innovación | PGC2018-099645-B-I00 | Xavier Trepat |
| Ministerio de Ciencia e Innovación | BFU2016-79916-P | Pere Roca-Cusachs |

The funders had no role in study design, data collection and interpretation, or the decision to submit the work for publication.

## Author contributions

Xarxa Quiroga, Conceptualization, Investigation, Visualization, Methodology, Writing – original draft; Nikhil Walani, Investigation, Methodology; Andrea Disanza, Investigation, Methodology, Writing – review and editing; Albert Chavero, Alexandra Mittens, Investigation; Francesc Tebar, Robert G Parton, María Isabel Geli, Methodology; Xavier Trepat, Funding acquisition, Methodology; Giorgio Scita, Funding acquisition, Methodology, Writing – review and editing; Marino Arroyo, Conceptualization, Funding acquisition, Methodology, Writing – review and editing; Anabel-Lise Le Roux, Conceptualization, Supervision, Investigation, Visualization, Methodology, Writing – original draft, Writing – review and editing; Pere Roca-Cusachs, Conceptualization, Supervision, Funding acquisition, Methodology, Writing – review and editing

## Author ORCIDs

Nikhil Walani ⓘ http://orcid.org/0000-0002-5248-9181
Xavier Trepat ⓘ http://orcid.org/0000-0002-7621-5214
Robert G Parton ⓘ http://orcid.org/0000-0002-7494-5248
María Isabel Geli ⓘ http://orcid.org/0000-0002-3452-6700
Giorgio Scita ⓘ http://orcid.org/0000-0001-7984-1889
Anabel-Lise Le Roux ⓘ http://orcid.org/0000-0003-4152-5658
Pere Roca-Cusachs ⓘ https://orcid.org/0000-0001-6947-961X

## Decision letter and Author response

Decision letter https://doi.org/10.7554/eLife.72316.sa1
Author response https://doi.org/10.7554/eLife.72316.sa2

# Additional files

## Supplementary files

• MDAR checklist

• Source code 1. Computational code for dynamics of single evagination.

## Data availability

Source data is provided for all figures in a corresponding source data file.

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

# Appendix 1

## Appendix 1—key resources table

| Reagent type (species) or resource | Designation | Source or reference | Identifiers | Additional information |
|---|---|---|---|---|
| Cell line (*Homo sapiens*) | Dermal fibroblast (normal, Adult) | Lonza | CC-2511 | Designated as NHDF in the paper |
| Cell line (*Mus musculus*) | WT MEF | Giorgio Scita Lab. DOIs: 10.1038/emboj.2013.208; 10.1016 /j.chom.2009.02.003 | | See first section of Materials and methods |
| Cell line (*M. musculus*) | IRSp53$^{-/-}$ MEF | Giorgio Scita Lab. DOIs: 10.1038/emboj.2013.208; 10.1016 /j.chom.2009.02.003; 10.1074/jbc.M808425200 | | See first section of Materials and methods |
| Cell line (*M. musculus*) | IRSp53$^{-/-R}$ MEF | Giorgio Scita Lab. DOIs: 10.1038/emboj.2013.208 | | See first section of Materials and methods |
| Cell line (*M. musculus*) | IRSp53$^{-/-}$ C/C9 IRTKS MEF | This paper, Giorgio Scita Lab. | | See first section of Materials and methods |
| Transfected construct (mouse) | siRNA to Nap1 (SMARTpool) | Dharmacon | ON-TARGETplus Mouse Nackp1 50884 | 50 nM |
| Transfected construct (mouse) | siRNA non targeting #3 (SMARTpool) | Dharmacon | ON-TARGETplus Non-targeting #3 | 50 nM |
| Antibody | Anti-BAIAP2, anti-IRSp53 (Rabbit polyclonal) | Sigma-Aldrich | HPA023310 | WB 1:200, used in Author response image 1 |
| Antibody | Anti-IRSp53 (Mouse monoclonal) | Giorgio Scita Lab 10.1038/emboj.2013.208 | | WB 5 µg/ml, used in *Figure 2—figure supplement 1* |
| Antibody | Anti-IRTKS (Mouse polyclonal) | Jan Faix Lab 10.1073/pnas.221743712 | | WB (1:500); used in *Figure 2—figure supplement 1* |
| Recombinant DNA reagent | mEmerald-Ezrin (plasmid) | Addgene | #54090 | |
| Recombinant DNA reagent | Actin-Chromobody plasmid (TagRFP) (plasmid) | Chromotek | Proteintech ACR | |
| Recombinant DNA reagent | mKate2-P2A-APEX2-csGBP (plasmid) | Addgene | #108875 | |
| Recombinant DNA reagent | pC1-EGFP-IRSP53FL (plasmid) | Giorgio Scita Lab 49 10.1038/ncb1502; 10.1038/s41467-020-17091-x | | |
| Recombinant DNA reagent | pC1-EGFP-IRSP53-4KE (plasmid) | Giorgio Scita Lab 43 10.1038/s41467-018-03955-w; 10.1038/s41467-020-17091-x | | |
| Recombinant DNA reagent | pC1-EGFP-IRSP53-I268N (plasmid) | Giorgio Scita Lab 43 10.1038/s41467-018-03955-w; 10.1038/s41467-020-17091-x | | |
| Recombinant DNA reagent | pC1-EGFP-IRSP53-I403P (plasmid) | Scita Lab 38 10.1038/emboj.2013.208: 10.1038/s41467-020-17091-x | | |
| Recombinant DNA reagent | pC1-EGFP-IRSP53-W413G (plasmid) | Scita Lab 38 10.1038/emboj.2013.208 | | |
| Recombinant DNA reagent | pC1-EGFP-IRSP53-IBAR (plasmid) | Scita Lab 38 10.1038/emboj.2013.208 | | |
| Recombinant DNA reagent | pC1-EGFP-IRSP53-ΔIBAR (plasmid) | Scita Lab 38 10.1038/emboj.2013.208 | | |

*Appendix 1 Continued on next page*

*Appendix 1 Continued*

| Reagent type (species) or resource | Designation | Source or reference | Identifiers | Additional information |
|---|---|---|---|---|
| Recombinant DNA reagent | mCherry-Mem | Tebar Lab 10.1111 /j.1600–0854.2011.01274 .x | | |
| Recombinant DNA reagent | Rac1-G12V | Tebar Lab Lab 10.1038/s41598-017-07130-x | | |
| Recombinant DNA reagent | Rac1-T17N | Tebar Lab Lab 10.1038/s41598-017-07130-x | | |
| Recombinant DNA reagent | pEGFP-C3 Wave2 | Klemens Rottner Lab | | |
| Recombinant DNA reagent | pSpCas9(BB)–2A-Puro(PX459)V2.0 | Addgene | 62988 | |
| Sequence-based reagent | Nap1_F, NCKAP1_F | Sigma, FM1_Nckap1 | PCR primers | CATTCGGGGCTACAATAAAC |
| Sequence-based reagent | Nap1_R, NCKAP1_R | Sigma BM1_Nckap1 | PCR primers | TTAGTGCAGACCGTAAAAAC |
| Sequence-based reagent | GAPDH_F | 10.1038/s41556-022-00927-7 | PCR primers | ATCCTGCACCACCAACTGCT |
| Sequence-based reagent | GAPDH_R | 10.1038/s41556-022-00927-7 | PCR primers | GGGCCATCCACAGTCTTCTG |
| Sequence-based reagent | Rn18x_F | This paper | PCR primers | GCAATTATTCCCCATGAACG |
| Sequence-based reagent | Rn18x_R | This paper | PCR primers | GGCCTCACTAAACCATCCAA |
| Sequence-based reagent | IRSp53_F, BAIAP2_F | Thermo Fisher | PCR primers ID: mm00499943_m1 | |
| Sequence-based reagent | IRSp53_R, BAIAP2_R | Thermo Fisher | PCR primers ID: mm00499943_m1 | |
| Sequence-based reagent | IRTKS_F, BAIAP2L1_F | Thermo Fisher | PCR primers ID: mm00508802_m1 | |
| Sequence-based reagent | IRTKS_R, BAIAP2L1_R | Thermo Fisher | PCR primers ID: mm00508802_m1 | |
| Sequence-based reagent | Pinkbar_F, BAIAP2L2_F | Thermo Fisher | PCR primers ID: mm00616958_m1 | |
| Sequence-based reagent | Pinkbar_R, BAIAP2L2_R | Thermo Fisher | PCR primers ID: mm00616958_m1 | |
| Sequence-based reagent | MIM_F, MTSS1_F | Thermo Fisher | PCR primers ID: mm00460614_m1 | |
| Sequence-based reagent | MIM_R, MTSS1_R | Thermo Fisher | PCR primers ID: mm00460614_m1 | |
| Sequence-based reagent | ABBA_F, MTSS1L_F | Thermo Fisher | PCR primers ID: mm01244296_m1 | |
| Sequence-based reagent | ABBA_R, MTSS1L_R | Thermo Fisher | PCR primers ID: mm01244296_m1 | |
| Sequence-based reagent | IRTKs sgRNA | This paper | sgRNA | AAAAGCCTACTACGACGGCG |
| Commercial assay or kit | RNAeasy kit | QIAGEN | 50974104 | |
| Commercial assay or kit | High Pure RNA Isolation Kit | Roche | 11828665001 | |
| Commercial assay or kit | Fast SYBR Green MasterMix | Thermo Fisher/Applied Biosystems | | |

*Appendix 1 Continued on next page*

Appendix 1 Continued

| Reagent type (species) or resource | Designation | Source or reference | Identifiers | Additional information |
|---|---|---|---|---|
| Chemical compound, drug | DAB tablet | Sigma-Aldrich | Sigmafast, D4293 | |
| Chemical compound, drug | PNB | Optopharma | DR-N-111 | |
| Chemical compound, drug | SMIFH2 | Abcam | ab218296 | |
| Chemical compound, drug | Wiskostatin | Sigma | W2270 | |
| Chemical compound, drug | CK666 | Merck | 182515 | |
| Software, algorithm | ImageJ | ImageJ | | freeware |
| Software, algorithm | GraphPad Prism 9 | GraphPad Prism 9 | GraphPad Prism 9.5.1.733 | |
| Software, algorithm | CellProfiler | CellProfiler | CellProfiler 4.2.4 | freeware |
| Software, algorithm | MATLAB | Marino Arroyo lab 10.1103/PhysRevLett.110.028101; 10.1088/1367–2630/ab3ad6, | This paper | Newly created MATLAB code, combined with previously existing code |

