## [Editor Report]

In this important paper, the authors characterize and model a novel self-organizing circuit used by cells to correct for nanoscale membrane evaginations that form upon a rapid drop in membrane area. The evidence supporting the conclusions is compelling and provides new insights on plasma membrane-cortex coupling. It is of interest to mechanobiologists and biophysicists and could be relevant in processes in developmental biology or in tissues where the cell area changes rapidly, such as in muscles, heart, and lungs.

---

## [Decision Letter]

**Decision letter after peer review:**

Thank you for submitting your article "A mechanosensing mechanism mediated by IRSp53 controls plasma membrane shape homeostasis at the nanoscale" for consideration by *eLife*. Your article has been reviewed by 3 peer reviewers, including X as the Reviewing Editor and Reviewer #1, and the evaluation has been overseen by Suzanne Pfeffer as the Senior Editor. The following individual involved in review of your submission has agreed to reveal their identity: Milos Galic (Reviewer #3).

Essential revisions:

The reviewers have considered that your work represents a well-executed study in a very defined reductionist model system. It provides new insights on plasma membrane-cortex coupling. After discussion, we think that your manuscript would nevertheless benefit from the following revisions:

1) Strengthen the model:

One issue of the paper is that the authors do not provide test of their model other than the absence of effect of Myo2 and do not discuss other mechanisms. The authors consider that the resorption is essentially related to actin production at the place where IRSp53 is localized, which induces an actin flow parallel to the plasma membrane, away from the evagination. Practically, this mechanism should also be present in any membrane protrusion, including filopodia. Thus, it would be important to compare the different actin fluxes at the evagination to determine the dominant contribution. What about the retrograde flow (perpendicular to the flow considered here) which would also lead to a resorption, and the protrusive forces that would counteract them? The authors mention the protrusive forces but do not explain why they are not strong enough.

2) Probe for additional proteins at the evagination sites

2.1. Other I-BAR proteins:

The resorption effect does not disappear in the absence of IRSp53 (Figures2H, 2L). It becomes just slower (Figure 2H), and actin is recruited but less (Figure 2L). Considering their redundant function, additional I-BAR domain proteins may contribute to this mechanism as well. To address this possibility, the authors should monitor the localization of other candidate proteins (e.g. MTSS1, MTSS2, IRTKS) at membrane evagination. Optionally, protein function could be tested via knockdown to delineate the contribution of individual I-BAR domain proteins.

2.2. Other N-BAR proteins:

It was shown for filopodia that N-BAR domain proteins limit their growth (See M. Galic et al., *eLife* 3 (2014)). Could it be the case here too?

2.3. Actin dynamics

Although it seems likely that IRSP3 triggers Rac and then Arp2/3 via WAVE, this is not supported by direct evidence. Moreover, it is not clear that this happens really locally or if global effects contribute. Hence it is important to localize Arp2/3 and components of the WAVE complex at the evagination sites. It is also important to knock out or knock down the WAVE complex to nail down the importance in this context.

Next, if they confirm their hypothesis, the authors could also comment on why the machinery for protrusion growth is not active at these sites, for instance why formins are not recruited and only Arp2/3 dependent polymerization takes place.

2.4. Clathrin

The process lasts over 3 min for the full resorption. This leaves time for other mechanisms that could influence lipid membrane total area, thus its tension (clathrin-independent endocytosis for instance that depends on actin), and lead to resorption of the bud. This is not considered in the paper and it the authors should test whether endocytosis plays a role.

3) Better characterization of phenotype:

3.1. Apical vs basal:

The phenomenon seems to occur entirely on the apical side as claimed in Figure 1, but the authors do not show what happens on the ventral part. The authors should clarify this point and provides evidence that the dots in the fluorescence image in Figure 1 are all apical.

3.2. Resorption time versus characteristic time

The authors mention that they cannot always measure the characteristic time of the evaginations flattening, and thus measure their time of resorption. But it becomes more difficult to compare quantitatively between different conditions. In particular, when they compare the time of resorption for different constructs, since this time might depend on the level of transfection/expression of the different IRSp53 constructs, the authors should normalize their data by the expression levels. Moreover, if the initial bud size varies, the full resorption will take longer even with the same decay time. This seems to be the case when comparing data with and without IRSp53 (Supp. Figure 1C-D and Figure 2H): the decay time seems similar but with a higher excess area at time zero in the absence of IRSp53 (higher PM signal) (also Figure 2H) which would correspond to larger buds. Thus, comparing the full duration of the processes rather than the characteristic time is probably not appropriate.

---

## [Author Response]

Essential revisions:The reviewers have considered that your work represents a well-executed study in a very defined reductionist model system. It provides new insights on plasma membrane-cortex coupling. After discussion, we think that your manuscript would nevertheless benefit from the following revisions:1) Strengthen the model:One issue of the paper is that the authors do not provide test of their model other than the absence of effect of Myo2 and do not discuss other mechanisms. The authors consider that the resorption is essentially related to actin production at the place where IRSp53 is localized, which induces an actin flow parallel to the plasma membrane, away from the evagination. Practically, this mechanism should also be present in any membrane protrusion, including filopodia. Thus, it would be important to compare the different actin fluxes at the evagination to determine the dominant contribution. What about the retrograde flow (perpendicular to the flow considered here) which would also lead to a resorption, and the protrusive forces that would counteract them? The authors mention the protrusive forces but do not explain why they are not strong enough.

We agree with the referees that our model in the original submission focused on the novel mechanism proposed to explain flattening of an evagination as a result of localized Arp2/3 actin polymerization. The literature amply discusses how actin polymerization exerts out-of-plane forces on the membrane, and hence favors membrane projections, whereas our experiments show that localized polymerization drives flattening. This motivated our proposed mechanism, based on a lateral gradient of cortical tension driving centrifugal flows. We agree that this mechanism is quite generic, and may also be present in other situations where localized polymerization drives membrane extensions. We note that the rationale of our model applies to Arp2/3 polymerization, and not to formin-based polymerization typical in filopodia, since Arp2/3 polymerization can push the membrane outwards, but can also compete with the contractile component of the actin network to locally reduce contractility.

However, the question of what controls the competition between out-of-plane protrusive forces and the proposed in-plane flattening mechanism remains.

Following the suggestions of the referees, in the revised manuscript we have extended our model. The reviewers mention retrograde flows and protrusive force. Regarding retrograde flows, we note that their length scales are larger than the 100 nm scale of our invaginations, and therefore could potentially displace but not likely reshape evaginations. Consistently, we did not observe any appreciable differences in the behavior of evaginations in the cell center, lamella, or lamellipodia, where retrograde flows should be very different. We therefore focused our model extension on protrusive forces. In our extended model, we accounted not only for the lateral tension gradients driving centrifugal actin flows that flatten the membrane, but also for out-of-plane protrusive polymerization forces. Based on quantitative in-vitro measurements of the relation between polymerization velocity and opposing pressure, we introduced a growing network perpendicular to the plane (coarse-grained as a filament at the center of the evagination), which upon contact with the membrane exerts a normal force on it, which can slow-down, stall, or reverse its growth velocity.

For reasonable parameter estimations, we find that this force is not able to stop the resorption and recovery of membrane/cortex homeostasis, but it slows down significantly the process. In our previous version of the model, resorption of the evagination was very abrupt (much faster than in experiments) and related to a mechanical snapping of the membrane bud under tension. With the modified model, snapping is limited by the disassembly of the out-of-plane network under membrane force, and instead we find a progressive resorption that better agrees with the experiments.

In the revised manuscript, figures and Methods section, we describe and motivate this more complete version of the model. We further consider two alternative scenarios to better appreciate individually the lateral and out-of-plane mechanical effects of localized polymerization. The first one corresponds to the model in our original submission, which leads to homeostasis but in an unrealistically abrupt manner. The second one considers only the out-of-plane part of the model. We show that depending on the strength of polymerization against the membrane, which can depend on a variety of actin regulators and cross-linkers, out-of-plane localized polymerization can result in a stable evagination or in its extension.

2) Probe for additional proteins at the evagination sites2.1. Other I-BAR proteins:The resorption effect does not disappear in the absence of IRSp53 (Figures2H, 2L). It becomes just slower (Figure 2H), and actin is recruited but less (Figure 2L). Considering their redundant function, additional I-BAR domain proteins may contribute to this mechanism as well. To address this possibility, the authors should monitor the localization of other candidate proteins (e.g. MTSS1, MTSS2, IRTKS) at membrane evagination. Optionally, protein function could be tested via knockdown to delineate the contribution of individual I-BAR domain proteins.

As the reviewers point out, the resorption effect is not total in the curves we show of the IRSp53 KO cell line, compared with the control cells reconstituted with IRSp53 proteins. As the reviewer suggested, this may come from the fact that I-BAR domain proteins may have redundant functions in this process. We verified the expression levels of the different proteins from the IBAR-domain family in the KO and reconstituted cell lines using RT-PCR. This revealed that IRTKs and MIM proteins were overexpressed (by 10 and 30 times fold respectively) in the IRSp53 KO cell line, and this overexpression was conserved in the reconstituted control (Author response image 1 or Figure 2—figure supplement 1H).

**Author response image 1. sa2fig1:** (A) RT-PCR from cell lysates of WT, IRSP53^-/-^, IRSP53^-/-^ silenced with IRTKs using CRISPR-Cas9 (clone F3) (B) Western blot image from cell lysates incubated with anti-IRSp53 antibody of WT, IRSP53^-/-^, IRSP53^-/-R^ from the different batches of MEF cells used. Representative images of 3 experiments. (C) Quantification dynamics of mCherry-membrane tagged PM evaginations after stretch release in WT or IRSp53^-/-^ MEF cells. N = 18 and 17 cells from 2 independent experiments. (D) Corresponding quantification of the decay constants of WT or IRSp53^-/-^ MEF cells. Significance was calculated through Mann-Whitney test. N = 20 and 17 cells from 2 independent experiments. (E) Western blot image of WT, IRSP53^-/-^ cell lysates used in C, D. Representative western blot of 3 experiments. (F) Quantification dynamics of mCherry-membrane tagged PM evaginations after stretch release in NHDF stable cell lines transfected with non-targeting shRNA or IRSp53-shRNA. N = 10 cells in each case. (G) Corresponding quantification of the decay constants of NHDF stable cell lines transfected with non-targeting shRNA or IRSp53-shRNA. Significance was calculated through unpaired t-test. N = 10 in each case. (H) Quantification dynamics of mCherry-membrane tagged PM evaginations after stretch release in IRSp53^-/-^ or IRSp53^-/-R^ MEF cells. N = 15 and 19 cells from 3 independent experiments. (I) Corresponding quantification of the decay constants of IRSp53^-/-^ or IRSp53^-/-R^ MEF cells. Significance was calculated through Mann-Whitney test. N = 15 and 19 cells from 3 independent experiments. Data show mean ± s.e.m.

We scrutinize the role of IRTKS, as its domain structure and protein function are very similar to those of IRSp53. For this, we used CRISPR/CAS9 to derive an IRTKS KO cell line in the IRSp53 KO MEFs background. Neither IRSp53 nor IRTKs were present in these cell lines as demonstrated by RT-PCR and WB. We monitored the resorption of the evaginations with time and compared WT and double KO MEF cells. Of note and as suggested by the reviewers and detailed in Response 3.2, we modified the quantification method. Instead of comparing fluorescence levels, we fitted curves to an exponential decay to compare the decay constants between conditions, which is what we now show in Figure 2—figure supplement 1I, J. Surprisingly, we found a slightly faster resorption in the double KO mutant versus the WT. We therefore repeated the experiments with the single IRSp53 KO, and did not find a significant difference (Figure 2—figure supplement1D, E or Author response image 1). Clearly, this did not seem consistent with our previous results, and prompted us to revisit our previous experiments using IRSp53 shRNA or KO. Regarding our shRNA measurements, analyzing decay times as suggested by the reviewers (rather than fluorescence levels) showed no difference between conditions (Author response image 1). Regarding our KO measurements, we realized through western blotting that we had mislabeled one batch of cell vials, and that in fact IRSp53 KO and reconstituted cells were inverted with respect to their label (Author response image 1). This affects the results that we reported in figures 2, 3 and 4 of the manuscript (but not Figure 5). As the reviewers will imagine, we were horrified by this, and we checked thoroughly our cells, in both the Roca-Cusachs and the Scita laboratories. By this we confirmed that this problem only and exclusively concerned the above-mentioned results. In any case, re-analyzing the results by computing decay times reduced somewhat the differences, but still led to significant differences between KO and reconstituted cells (Author response image 1). However, with the correct label, the differences mean that KO cells resorbed evaginations faster, not slower, than reconstituted cells. We apologize for this serious error, which we hope to correct by thoroughly revising the manuscript.

Overall, our results therefore showed inconsistent results of IRSp53 depletion or deletion, combined with an important overexpression of related I-BAR proteins. Given the potential compensatory role of these related proteins, we therefore conclude that knock-down or knock-out experiments are not well-suited to address our question. However, we note that our experiments with transient transfection of IRSp53 mutants (which will minimize overexpression problems) remain valid. After correcting our mislabeling, these experiments were carried out in IRSp53-reconstituted rather than KO cells, but the conclusions here remain valid. First, overexpressing WT IRSp53 has no effect, thereby showing that transient overexpression per se does not affect membrane resorption. Then, overexpression of mutant IRSp53 protein impairs resorption, likely through a dominant-negative rather than rescue effect. We have verified that the effect on resorption holds when decay times rather than fluorescence intensities are calculated. Further, the effect on decay times still correlates with the delay in protein recruitment, as we reported in our first submission.

Apart from these results with transient mutant transfection, we have also verified localization of IRSp53 using the Apex technique and confirmed its enrichment in the evagination. Our results also show the role of Rac1 and Wave Regulatory Complex (WRC) in the resorption process (we now have given stronger evidence in the revised manuscript as explained in the Response to point 2.3). The synergy in IRSP53, Rac1 and WRC interactions leading to Arp2/3-mediated actin polymerization is very well described. Therefore, we think that our data show that IRSp53 participates in recognizing the curved structure of the evaginations, leading to WRC recruitment in a feedback loop promoting actin polymerization. After this revision, it is clear that IRTKs and MIM proteins may play similar or compensatory roles, and therefore the role of IRSp53 may not be unique. We have toned down the manuscript conclusions accordingly. We have also modified substantially Figure 2 and 3 to update for these changes, and introduced Figure 2—figure supplement 1 and Figure 3—figure supplement 2 with additional new results. We have also modified the text accordingly.

2.2. Other N-BAR proteins:It was shown for filopodia that N-BAR domain proteins limit their growth (See M. Galic et al., eLife 3 (2014)). Could it be the case here too?

This is an interesting possibility. In the paper by Galic et al., an N-BAR protein is recruited through curvature sensing at the neck of nascent filipodia, which arise at previously formed actin patches. The negative curvature is generated through acto-myosin contraction from these patches, counterbalancing the protruding forces necessary for filipodia extension. The N-BAR protein acts as a Rac1 inhibitor therefore limiting growth of the filipodia. The mechanism described is very different from our findings, in which evaginations are passively formed at stochastic places of the plasma membrane to accommodate excess area. In our case, Rac1 activation promotes the flattening rather than growth of the evagination, so we do not think that this N-BAR-mediated mechanism can explain the response. Further, in an initial screening that we carried out at the beginning of this project, results of our preliminary experiments overexpressing some N-BAR or F-BAR domain containing proteins (Amphiphysin, FCHo1) did not show a slower resorption. We did not see either a faster resorption in stable cell lines where we silenced Amphyphisin, FCHO1, or GRAF1. Of course, we could test more proteins, and there could potentially be compensation mechanisms, such as the one described for IRSp53 above. However, in the manuscript we preferred to focus on I-BAR domain containing proteins and subsequent actin nucleating factor recruitment, due to their potential role in mediating the restoration of membrane homeostasis.

To better clarify this issue, we now mention the potential contribution of other BAR proteins in the discussion of the manuscript (page 9).

2.3. Actin dynamicsAlthough it seems likely that IRSP3 triggers Rac and then Arp2/3 via WAVE, this is not supported by direct evidence. Moreover, it is not clear that this happens really locally or if global effects contribute. Hence it is important to localize Arp2/3 and components of the WAVE complex at the evagination sites. It is also important to knock out or knock down the WAVE complex to nail down the importance in this context.

As the reviewer asked, we have studied the localization of the Wave complex components at the evagination. This kind of study is not straightforward given that any overexpressed, membrane-bound fluorescent protein will show an increase in signal at the evagination due to the formation of a 3D-shape membrane. For IRSp53 we have used the Apex technique, which was very complicated and low yield. Therefore, for the Wave complex, we decided to overexpress pEGFP-C3 Wave2 (the construct that gave best transfection results among all the plasmids available to us) and compared its enrichment relative to the plasma membrane fluorescent marker used for membrane labelling throughout the study. For this, we fixed the cells just after stretch release, and imaged then in Airyscan mode, which has improved resolution compared with classical confocal microscopy. To make sure we were restricting the analysis to evaginations, we prepared the same samples for electron microscopy imaging and used, as previously done, a pattern on the PDMS substrate to obtain SEM images of the same cells. For the structures clearly identified as evaginations in the SEM image, we calculated the ratio between the fluorescence intensity in the evagination, and in an adjacent area containing only flat membrane. This ratio was significantly higher for the WAVE-C-GFP signal than for the membrane signal, (Figure 5A, B), which confirms enrichment of the complex at the evagination.

We also proceeded to silence the Nap1 protein from the Wave complex (Nap1 is present in both Wave1 and Wave2 complexes) using siRNA. We subjected cells to 50nM siRNA targeting the Nap1 gene and using RT-PCR, we saw a robust decrease of Nap1 RNA levels when compared with 50nM of non-targeting siRNA control. We initially had the idea of co-transfecting the siRNA together with the plasma membrane marker to perform live experiments, unfortunately the yield of the fluorescence labelling in this context was extremely low. Therefore, we opted for an alternative experiment employing SEM imaging at a fixed time, as the images obtained by SEM very clearly show the evaginations without the need for fluorescence labelling. We fixed the sample 25 s after the release of the stretch to allow for some resorption to occur and compared the images of cells where Nap1 was silenced compared with the non-targeting siRNA control. We quantified the number of evaginations per µm^2^ of cell membrane. Cells silenced for Nap1 (thereby severing Wave complex functions) showed a significantly higher amount of evaginations (Figure 5C-E), indicating that the resorption was impaired under this condition.

Overall, we believe that the use of CK666 to inhibit Arp2/3 (which clearly showed impeded resorption) combined with this new evidence of WAVE localization and function at the evagination convincingly show that the actin polymerization event observed at the evagination is driven by this molecular cascade.

We have now inserted these results in Figure 5 and Figure 5—figure supplement 1, and their corresponding explanation in the text (page 8).

Next, if they confirm their hypothesis, the authors could also comment on why the machinery for protrusion growth is not active at these sites, for instance why formins are not recruited and only Arp2/3 dependent polymerization takes place.

This is an interesting question. We have shown in the past (Disanza et al. EMBO J, 2013) that mDia is not recruited at the Cdc42/VASP/IRSp53 machinery foci driving filopodia elongation in MEFs. This may be used to underline that different machinery may be recruited and function at different sites (not necessary in common) and that, at least, formins may be excluded by some IRSp53 driven actin regulatory processes. Additionally, treatment with the formin inhibitor SMIFH2 reduced the number of filopodia as expected but did not affect evagination resorption. These results are shown in Figure 4—figure supplement 1C and D. Supporting this notion, our modelling results show that membrane flattening can be explained by an isotropic actin flow, which would be mediated by branched rather than linear polymerization. We now comment on this fact in the discussion part of the revised manuscript.

2.4. ClathrinThe process lasts over 3 min for the full resorption. This leaves time for other mechanisms that could influence lipid membrane total area, thus its tension (clathrin-independent endocytosis for instance that depends on actin), and lead to resorption of the bud. This is not considered in the paper and it the authors should test whether endocytosis plays a role.

We agree with the reviewers that upon stretch release, endocytosis indeed occurs and plays a role in plasma membrane homeostasis. This was the central topic of one of our previous publications (Thottacherry et al., Nat. Com. 2018), where we characterized how membrane tension decrease upon stretch release affects CDC42-mediated CLIC-GEEC endocytosis (but not clathrin-mediated endocytosis). In the original submission of this manuscript, we acknowledged that endocytosis increases in response to decreased plasma membrane tension, and we cited the Thottacherry et al. study in our introduction. However, this cannot explain our results for two main reasons. First, the actin polymerization pathways involved are different (CDC42-mediated in the case of endocytosis, Wave-mediated in the case of evagination resorption). Second, endocytic buds do not colocalize with stretch-mediated evaginations, as also characterized by Thottacherry et al. We have now clarified this in the introduction and in the discussion (page 9), which we have modified with our revised findings.

3) Better characterization of phenotype:3.1. Apical vs basal:The phenomenon seems to occur entirely on the apical side as claimed in Figure 1, but the authors do not show what happens on the ventral part. The authors should clarify this point and provides evidence that the dots in the fluorescence image in Figure 1 are all apical.

We are aware, as the reviewers correctly point out, that we only show evaginations of the apical side of the cell. We have attempted to characterize what happens on the ventral part using electron microscopy as well but were not successful. Using TEM to prepare the samples, we must peel the cells from the PDMS membrane to proceed with serial cuts, but in this process the ventral membrane is torn. We have tried to unroof cells just after stretch release and perform SEM imaging, but the whole protocol was quite slow compared with the resorption time scale, and also rendered dirty samples that could not resolve the structure of basal invaginations. Still, alignment of the SEM images with the fluorescence stainings (Figure 1C) show that a large part of the fluorescence dots come from evaginations at the apical membrane. We do know that curved membrane structures appear at the ventral part as well (as per our confocal images in Kosmalska et al., Nat. Com. 2015, Figure 2c) and we hypothesize that invaginations will form at the ventral part case given the high rigidity of the PDMS substrate. We also hypothesize that in physiological situations, invaginations will emerge if it is more favorable for the membrane to push inward the cytoskeleton, which would be the case of very rigid and non-porous substrates. In many other cases, evaginations will emerge if is it more favorable to bend the membrane outwards, as happens here at the apical side of the cell. We only studied the machinery involved in the resorption of the evaginations, which in our case was the majority of the structures formed, and that we think are highly physiologically relevant. We now clarified this aspect in the introduction (page 2).

3.2. Resorption time versus characteristic timeThe authors mention that they cannot always measure the characteristic time of the evaginations flattening, and thus measure their time of resorption. But it becomes more difficult to compare quantitatively between different conditions. In particular, when they compare the time of resorption for different constructs, since this time might depend on the level of transfection/expression of the different IRSp53 constructs, the authors should normalize their data by the expression levels. Moreover, if the initial bud size varies, the full resorption will take longer even with the same decay time. This seems to be the case when comparing data with and without IRSp53 (Supp. Figure 1C-D and Figure 2H): the decay time seems similar but with a higher excess area at time zero in the absence of IRSp53 (higher PM signal) (also Figure 2H) which would correspond to larger buds. Thus, comparing the full duration of the processes rather than the characteristic time is probably not appropriate.

We thank the reviewer for insisting on that point and agree with them that the decay constant (inverse of the characteristic time, and obtained from an exponential fit) is more informative for our study. The reason for not measuring decay constants in our original submission was that fluorescence curves from individual reservoirs were often noisy, and not well fitted by an exponential decay curve. To circumvent this, we have now calculated an average decay curve for all the reservoirs within a given cell, and then fitted this average curve. In this way, noise is largely averaged out, and the curves are well fitted to an exponential decay. Overall, the fit does not drastically change the results, except for the shRNA data, as discussed in detail in response to reviewer question 2.1. All the data have now been fitted with an exponential decay curve and comparison of the decay constants have been used in the analysis.